# Multimodal imaging reveals a lysosomal drug reservoir that drives heterogeneous distribution of PARP inhibitors

Carmen R. Moncayo [1], Restuadi Restuadi [2,7], Guanying Zhang[1,7],
Daniel Marks [3], Paula Ortega-Prieto[1], Emily Doherty [2], Nathalie Lambie [4],
Chad Whilding[1], Ivan Andrew[1], Alex Montoya [1], Bhavik Patel[1], Katie Tyson [3],
Betheney R. Pennycook [2], Lauren Pendergast[5], Vincen Wu[6], Zoltan Takats [6],
Nik Matthews [4], George R. Young [1], Priyanka Verma [5], Pavel Shliaha [1],
Laurence Game[1], Boris Lenhard [1,2], Iain McNeish [3], Christina Fotopoulou[3],
Alexis R. Barr [1,2], Paula Cunnea [3], Zoe Hall [6] ✉ & Louise Fets [1,2] ✉

For all drugs, effective target engagement requires sufficient intracellular concentrations of drug to be reached, but whether tumour heterogeneity impacts drug distribution and efficacy is poorly studied. Poly (ADP-ribose) polymerase (PARP) inhibitors have transformed treatment opportunities for women with high-grade serous ovarian carcinoma, but resistance remains a clinical hurdle in this highly heterogeneous tumour type. Here, we present a patient-derived explant multi-modal imaging pipeline, which demonstrates that cell-intrinsic PARP inhibitor accumulation is highly variable, both between patients and within tumours. Spatial transcriptomics reveals enrichment of apoptotic and lysosomal signatures in high-drug regions. Rucaparib, an intrinsically fluorescent PARP inhibitor, accumulates heterogeneously at the single-cell level, with rucaparib-high cells demonstrating increased drug response relative to rucaparib-low. Mechanistically, lysosomal sequestration creates a rucaparib reservoir that determines drug levels in the nucleus. Perturbation of lysosomal content alters intracellular levels of weak base PARP inhibitors rucaparib and niraparib, but not olaparib. Together these data suggest that lysosomes act as a reservoir for a subset of PARP inhibitor drugs to improve drug response.

To be effective in tumours, therapeutic drugs must reach a concentration at which they can effectively bind their target, and the intracellular concentration achieved will also dictate off-target effects[1]. Though it is clear that tumour heterogeneity can drive therapeutic resistance[2], both at the inter-patient level and at the intra-tumoural level, whether tumour heterogeneity impacts upon drug accumulation

and distribution, and how this influences drug efficacy and resistance has yet to be explored.

High-grade serous ovarian carcinoma (HGSOC) is a particularly heterogeneous disease at the intra-patient level, both spatially and temporally[3,4]. Genetic heterogeneity is well described, and includes near-ubiquitous *TP53* mutations[5], substantial copy number variation

[1]MRC Laboratory of Medical Sciences, London, UK. [2]Institute of Clinical Sciences, Imperial College London, London, UK. [3]Department of Surgery and Cancer, Imperial College London, London, UK. [4]Imperial BRC Genomics Facility, Imperial College London, London, UK. [5]Division of Oncology, Washington University School of Medicine, St. Louis, MO, USA. [6]Department of Metabolism, Digestion and Reproduction, Imperial College London, London, UK. [7]These authors contributed equally: Restuadi Restuadi, Guanying Zhang. ✉e-mail: zoe.hall@imperial.ac.uk; l.fets@lms.mrc.ac.uk

and high-frequencies of homologous recombination DNA repair deficiency (HRD), most commonly through *BRCA1/2* mutation[6–8]. Non-genetic heterogeneity is also well documented in HGSOC, including within fallopian tube epithelia, the cell type of origin[9], and substantial phenotypic diversity has been observed in tumour cells from multiple patient cohorts[4,10–12].

Patients with HGSOC typically undergo cytoreductive surgery, combined with platinum-based chemotherapy. The introduction of additional targeted maintenance therapies such as PARP inhibitors has led to significantly improved remission rates in HGSOC patients[13,14]. PARP inhibitors exploit HRD and lead to synthetic lethality, with three agents in current clinical use for HGSOC: olaparib, niraparib and rucaparib. Mechanistically, these drugs cause PARP molecules to become trapped on DNA in addition to inhibiting their catalytic activity. This results in replication fork stalling and the formation of DNA double strand breaks (DSBs)[15], which ultimately lead to apoptosis of tumour cells.

Variability in PARP inhibitor accumulation, at the intra- or inter-patient level, has yet to be studied, despite evidence that fluorescent analogues of PARP inhibitors accumulate heterogeneously in in vitro and pre-clinical in vivo models[16]. Whether this is mimicked by the clinically used drugs is unknown, but could be of significant consequences if a subpopulation of cells accumulates drug to a level below that required for target engagement, which could contribute to resistance. Therefore, we set out to determine whether PARP inhibitors are heterogeneously accumulated in HGSOC and whether this impacts their efficacy.

In this work, using HGSOC patient-derived explant (PDE) models, we develop a pipeline to treat tumour slices with PARP inhibitors ex vivo. Mass spectrometry imaging (MSI) demonstrates spatial heterogeneity in drug distribution both between and within patient tumours. We use spatial transcriptomics of adjacent tissue sections to compare regions of high- and low-drug concentration, revealing that differential drug accumulation has significant effects on drug response and is associated with lysosomal signatures. Using established ovarian cancer cell lines, we demonstrate that heterogeneity in drug response occurs at the single-cell level, and that this is linked to cell-to-cell variability in drug accumulation. Mechanistically, we demonstrate that lysosomal sequestration of both rucaparib and niraparib, but not olaparib, drives increased uptake in high-drug cells. Together, these data suggest that heterogeneous drug distribution may be a contributing factor to drug resistance in a clinical setting.

## Results

### A spatially-resolved pipeline to assess drug distribution in PDEs

We hypothesised that the substantial genetic and non-genetic heterogeneity in HGSOC[3,10,11] may impact distribution of PARP inhibitors within tumours. To test whether cell-intrinsic properties could lead to differential drug accumulation, we required a model in which PARP inhibitors could be delivered to the tumour independently of the vasculature. To achieve this, we used a PDE model using surplus surgical tissue from treatment-naïve patients with HGSOC who had undergone maximal-effort primary cyto-reductive surgery. Samples from five tumours, derived from three patients, were each sectioned into multiple slices (details in Supplementary Data 1) and cultured ex vivo, enabling us to incubate slices from the same tumour with multiple drugs. There are a number of PARP inhibitors that have been tested in clinical trials with varying chemical properties (Supplementary Data 2). We chose to focus on olaparib, niraparib and rucaparib since these are licenced for use in HGSOC (Fig. 1a). Twenty one independent PDEs were processed using this pipeline in total. After treatment, each PDE slice was snap frozen, then cryo-sectioned into multiple 10 μm-thick sections to assess drug distribution, drug response and tissue morphology (Fig. 1b).

To visualise drug distribution in the treated-PDE tissue sections, we used mass spectrometry imaging (MSI), which showed excellent signal linearity with concentration for all three drugs tested (Supplementary Fig. 1a-c). Using olaparib-treated samples as a test-case, MSI was used to image drug distribution in slices treated for varying amounts of time, from which we determined that steady-state drug distribution was reached after 24 h of treatment (Supplementary Fig. 1d-f).

The distributions of niraparib, rucaparib and olaparib were compared in 18 PDEs cultured after tumour resection from three patients with HGSOC. One additional PDE from each patient was treated as a vehicle-only control to check the specificity of the MSI signal. PARP inhibitor signal was normalised relative to a heavy isotope-labelled analogue of each drug, homogeneously deposited across each tissue slice as an internal standard, to enable semi-quantitative comparison of drug accumulation between patients. Where tissue availability allowed, multiple slices were cultured for each patient/drug combination, and tumours were sourced from multiple sites (Supplementary Data 1). For all tissue slices, six 10 μm-thick sections were used for multi-modal imaging analyses. The section analysed using Matrix-Associated Laser Desorption Ionisation (MALDI) MSI, deemed section 0, was subsequently haematoxylin and eosin (H&E) stained to examine tissue architecture. The adjacent section, section +1, was used for GeoMx spatial transcriptomics, while immunohistochemical (IHC) staining of sections −2, −1, +2 and +3 probed apoptosis and DNA damage levels using cleaved caspase 3 and γH2AX, as well as HGSOC markers Wilms' tumour 1 (WT1) and Pax8. Supplementary Fig. 2 presents the results of these analyses in a grid form; specific comparisons will be referred to using the coordinates of this grid. FIGO stages of patients can be found in Supplementary Data 1.

### Cell-intrinsic heterogeneity in PARP inhibitor accumulation in HGSOC explants

Despite dosing independently of the vasculature system, a high degree of heterogeneity in PARP inhibitor distribution was observed across all patients (Fig. 1c, Supplementary Fig. 2 row 3), with clear drug hotspots in some parts of the tumour, while other regions accumulated little or no detectable drug. Interestingly, the uptake of the three PARP inhibitors varied between patients, with relative levels of niraparib and rucaparib being comparable in the different patients, while olaparib accumulation differed (Fig. 1c, d). For example, samples from Patient 1 accumulated little olaparib, whereas levels of niraparib and rucaparib were highest in these samples relative to the others tested. This suggests that the differential uptake of drugs between patients is not due to a general difference in small molecule permeability within these tumours, but that specific mechanisms may govern the uptake and/or efflux of rucaparib and niraparib when compared to olaparib.

Although some examples of this heterogeneity were associated with differences in tissue architecture by H&E staining (e.g., a correlation between drug intensity in regions of increased cell density in Patient 2 after treatment with niraparib or rucaparib, Supplementary Fig. 2, E3:4,N3:4), this was not always the case (e.g. Patient 1, niraparib, A3:4). Regions of high and low drug accumulation did not appear to correlate with levels of tumour markers such as Pax8 or WT1 (Supplementary Fig. 2, rows 3, 9 and 10), suggesting that the drug does not exclusively accumulate in cancer cells, and within the cancer cell populations, there is variability in accumulation. In a number of cases, there was a correlation between the drug distribution and the staining pattern of γH2AX, a readout of DNA damage levels (Supplementary Fig. 2b, rows 2 and 3).

Inter-tumour heterogeneity in drug accumulation was also observed between different tumours from the same patient (Fig. 1e, Supplementary Fig. 2 row 3). Niraparib-treated samples from patient 2 derived from the omentum (E) and left ovary (F–H), showed differing

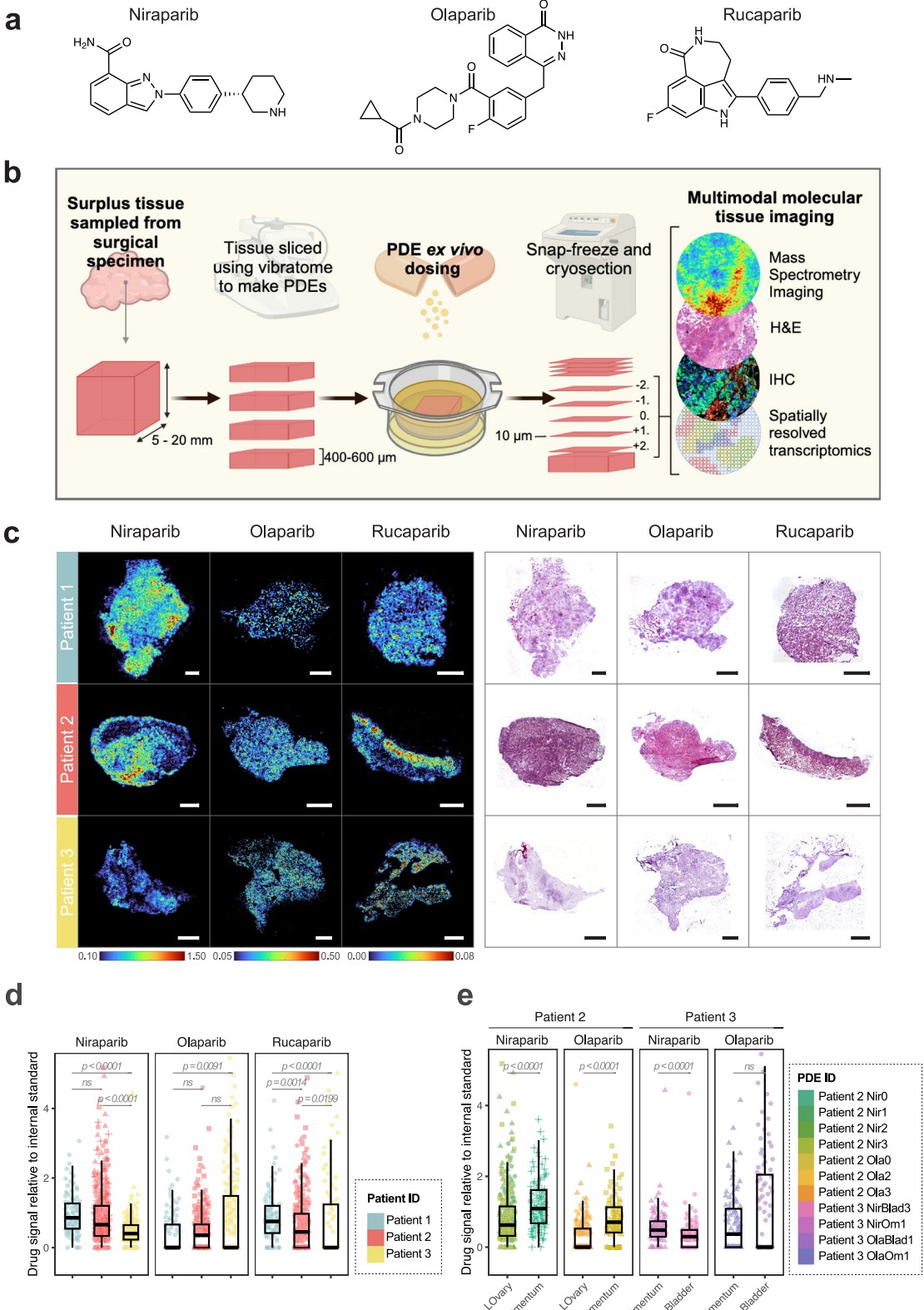

drug accumulation, as did olaparib-treated samples derived from the same sites (I and J, K respectively). Similarly, niraparib-treated samples derived from the bladder (P) and omentum (Q) of patient 3 displayed variability, as did olaparib-treated samples from these locations (R, S). Together, these data demonstrate that the accumulation of PARP inhibitors in patient-derived explants is highly heterogeneous, at the intra-tumour, inter-tumour and inter-patient level.

## Drug hotspots show significantly increased drug response signatures

To understand what drives the differential accumulation of PARP inhibitors and how this impacts their efficacy, we used the NanoString GeoMx spatial transcriptomics platform to determine whole-transcriptome differences between high- and low-drug regions of our PDEs. To achieve sufficient sequencing depth for whole-

**Fig. 1 | Mass Spectrometry Imaging reveals vasculature-independent inter- and intra-patient heterogeneity in PARP inhibitor distribution. a** Chemical structures of PARP inhibitor drugs used in this study. **b** A vasculature independent, patient-derived explant (PDE) pipeline to investigate mechanisms of drug accumulation, response and resistance in a spatially-resolved manner. H&E: haematoxylin and eosin; IHC: immunohistochemistry. *Created in BioRender. Ramirez moncayo, C. (2026)* https://BioRender.com/mh7g4wz. **c** Representative MALDI mass spectrometry images (MSI) of PARP inhibitor drug distribution in PDEs dosed ex vivo. Niraparib $[M + Na]^+$ ($343.1535 \pm 5$ ppm), olaparib $[M + Na]^+$ ($457.1652 \pm 5$ ppm) and rucaparib $[M + H]^+$ ($324.1512 \pm 10$ ppm) signals, relative to drug-matched internal standards, are shown on the left. Colour scales depict relative ion intensities. To the right, H&E stainings of the same tissue sections post-MSI. Scale bar = 1 mm. **d** Inter-patient heterogeneity in PARP inhibitor accumulation. Drug signal (per pixel, relative to matched internal standards) was normalised to the third quartile (Q3) of combined data from all samples within that drug, to enable plotting

of all three drugs on the same scale. To capture the degree of intra-patient spatial heterogeneity, individual PDE samples are represented with different shapes and tissue pixels were grouped into percentiles based on drug distribution, with each percentile represented as an individual data point. Two-sided Wilcoxon rank sum test with Bonferroni correction was applied across percentiles to compare differences in drug signal intensities across patients. **e** Inter-tumour heterogeneity in PARP inhibitor accumulation. Data were processed as described in (**d**). Statistical significance was assessed using a two-sided Wilcoxon rank-sum test across percentiles. No adjustments were made for multiple comparisons. For (**c**, **d**), Data are derived from $n = 3$ patients. Within each patient, data were collected from multiple PDEs: Patient 1 ($n = 1$), Patient 2 ($n = 2–4$), and Patient 3 ($n = 1–2$), depending on the drug condition. See Supplementary Data 1 for full details. Box plots represent the median (centre line), 25th and 75th percentiles (bounds of box), and minima and maxima (whiskers). ns: $p > 0.05$. Source data are provided as a Source Data file.

transcriptome analysis using this platform, a minimum number of nuclei is recommended per ROI. This was difficult to achieve in olaparib-treated samples as drug hotspots were, on average, much smaller than those observed in niraparib or rucaparib-treated samples (Fig. 1c, Supplementary Fig. 2, row 3), therefore we chose to focus only on the latter two drugs.

Regions of high- and low-drug were chosen based on MSI images of drug distribution and cross-referenced with the tissue morphology marker pan-cytokeratin and α-smooth muscle actin, to confirm that transcriptional profiles obtained were from cancer cells only and that fibroblasts were excluded (Fig. 2a, Supplementary Fig. 2, rows 5:8). Finally, regions were compared with IHC staining of ovarian cancer cell-specific markers, WT1 and Pax8, on alternative sections (Supplementary Fig. 2, rows 9:10). Twelve tissue sections were analysed, and an average of 5 high- and 5 low-drug regions of interest (ROIs) were obtained per tissue section. Quantification of the levels of drug between the ROIs confirmed a significant increase in drug levels in high ROIs relative to low for both niraparib and rucaparib (Fig. 2b).

After QC and filtering steps, we were left with 122 ROIs, with at least 20 regions of high- and low-drug for each of the two PARP inhibitors (Supplementary Fig. 3a), and across these, 11224 targets were reliably quantifiable. PCA and Euclidean distance clustering analysis demonstrated that, as expected, the major driver of transcriptional variance was inter-patient heterogeneity (Supplementary Fig. 3b, c). Within each patient, samples grouped by compound, then by PDE slice ID, and within each slice, high and low drug content areas were separated from one another, demonstrating that differential drug accumulation was associated with divergent transcriptomes (Supplementary Fig. 3b). Pathway analysis of all high versus low regions using the Reactome database demonstrated that high-drug regions were enriched in terms associated with cell cycle arrest and DNA damage response ('G1/S Damage Checkpoint'), and apoptosis ('Apoptosis', 'Apoptotic execution phase', 'Programmed cell death', 'Apoptotic cleavage of cellular proteins'), suggesting that PARP inhibitor response was enhanced by the increased concentration of the drugs, driving increased apoptotic signalling (Fig. 2c, Supplementary Data 3).

### High drug accumulation is associated with lysosomal gene signatures

By incorporating the relative drug concentrations quantified across each ROI (Fig. 2b), we were able to use drug level as a continuous (rather than a binary high versus low) variable as part of a separate linear mixed model for each of our two drugs used, with nested random effects to account for patient and PDE slice-derived variability. Using this, we identified genes whose expression was significantly associated with increasing or decreasing levels of either rucaparib or niraparib (Supplementary Fig. 4a, b, Supplementary Data 4). Enrichment analysis identified several overlapping terms associated with increasing levels of both drugs, centring around proteolysis or peptide

activity, extracellular matrix features and glycolytic activity (Supplementary Fig. 4c, Supplementary Data 5).

Though far more significant gene associations were observed with increasing niraparib concentration (2541, <0.05 FDR), 55% of genes that were associated with increasing rucaparib concentration (201, <0.05 FDR) were also found to be associated to niraparib (111 labelled gene transcripts, Fig. 2d, Supplementary Data 4). Strikingly, the effect sizes of transcripts that were significantly associated with both drugs were highly correlated (Fig. 2d, *slope* = 0.94, $R^2$ = 0.73).

Among the transcripts most strongly associated with increasing concentrations of niraparib and rucaparib were *CTSD*, encoding the lysosomal protease cathepsin D; the complement component *C3*, which has been demonstrated to be found in lysosomes in several cell types[17,18]; and *CMTM6*, the protein product of which has been shown to stabilise PD-L1 at the plasma membrane by promoting its sorting towards recycling endosomes and away from lysosomes[19,20] (Fig. 2d, relevant genes highlighted in red). Mirroring these associations with the endosome/lysosome system, GO term Cellular Component enrichment analysis of the genes commonly associated with both drugs revealed that ten of the top twenty terms referred to lysosomes or lysosome-related organelles (e.g. azurophilic granules, lytic vacuoles) (Fig. 2e, Supplementary Data 6).

On closer inspection 13 of the 111 common gene transcripts encoded lysosome-associated proteins (Fig. 2f), and the correlation between levels of drug and level of transcript expression was striking for well-known lysosomal genes such as *LAMP1*, *CTSD*, *APP* and *GRN* (Fig. 2g, Supplementary Fig. 4d). We also noted that 'Neutrophil Degranulation', a process known to be reliant on the secretion of lysosomal proteases, was also enriched in our binary analysis (Fig. 2c). Together, these data suggest that increased accumulation of both rucaparib and niraparib are associated with lysosomal signatures in PDEs.

### Single cell heterogeneity in PARP inhibitor-induced DNA damage response

PDE models capture HGSOC tumour heterogeneity and cellular diversity well, however 24 h treatment times are required to reach steady state levels of drug in this system (Supplementary Fig. 1d–f). This prolonged PARP inhibitor exposure prevents the deconvolution of features which drive heterogeneous drug accumulation from those features that arise as a result of concentration-dependent differences in drug response. We therefore set out to establish a cell-based model to better understand whether high-drug-associated lysosomal signatures could be a cause or a consequence of differential PARP inhibitor accumulation.

To determine whether we could model heterogeneous PARP inhibitor accumulation and response in cell lines, we first examined response to olaparib, rucaparib and niraparib at the single-cell level in the established HGSOC cell line, PEO1, as well as other ovarian cancer (OVCA) cell lines, using accumulation of $\gamma$H2AX as a read out for DSBs (Fig. 3a). Dosing at their $IC_{50}$ concentration (Supplementary Fig. 5a, b),

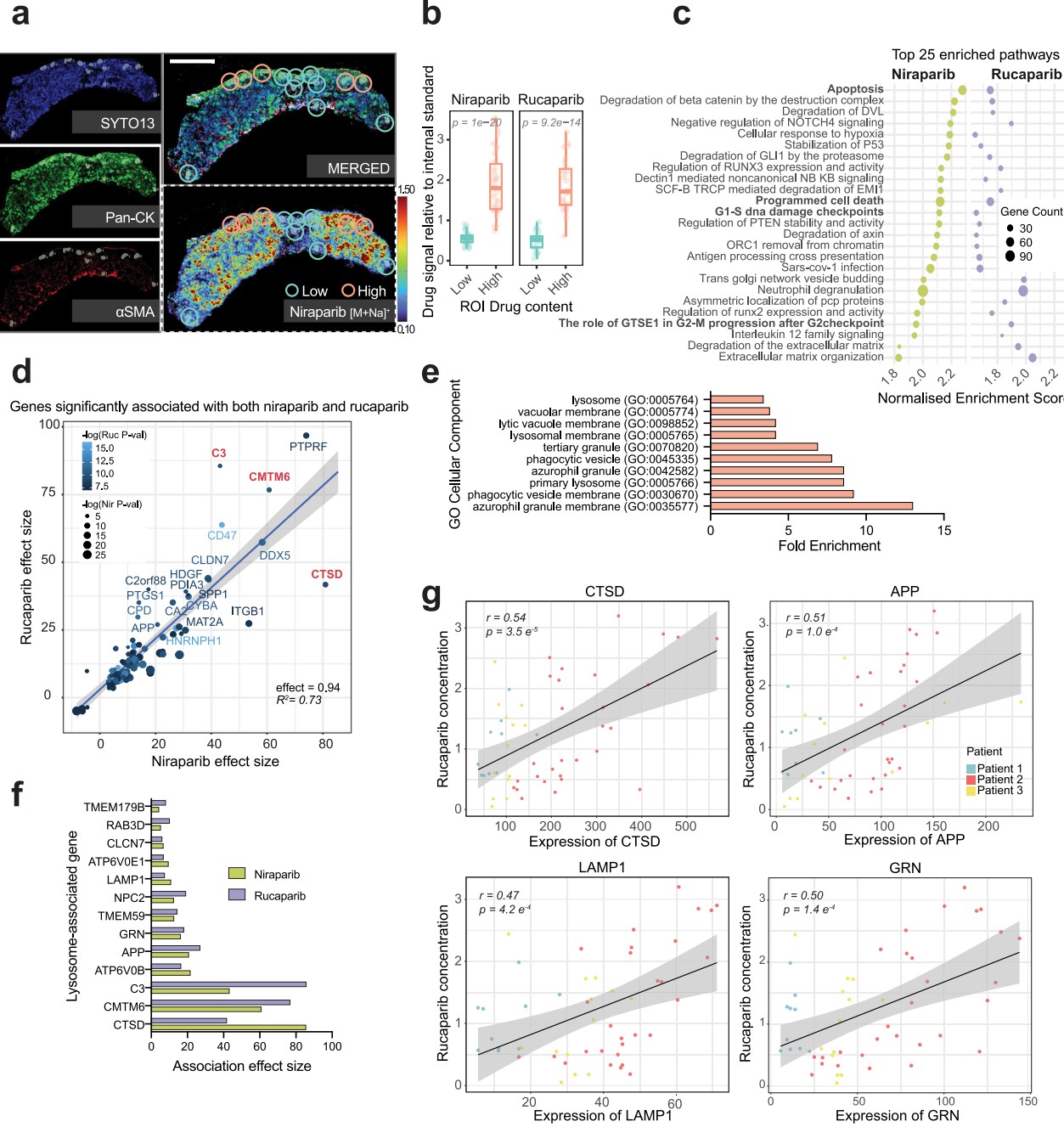

**Fig. 2 | Regions of high PARP inhibitor accumulation show increased apoptotic response and lysosomal signatures. a** Example of high and low region of interest (ROI) selection for GeoMx spatial transcriptomics in patient-derived explants (PDEs) following ex vivo dosing with PARP inhibitors. The MSI image (dashed bottom-right panel), displaying niraparib distribution, was used to inform ROI selection on panCK-positive, αSMA-negative tumour areas. Colour scale depicts relative drug intensity. Scale bar = 1 mm. **b** ROI drug quantification, relative to internal standards spiked in matrix, and Q3-normalised (as in Fig. 1d) to enable same-scale plotting. Data were derived from $n = 3$ biological replicates (patients). For niraparib and rucaparib respectively, $n = 38/40$ and $n = 27/26$ high/low ROIs were collected. Box plots represent the median with interquartile range (bounds of box), and minima and maxima (whiskers). Two-sided Wilcoxon rank sum test was applied to evaluate differences in drug intensity signal between low and high ROIs. **c** Top 25 Reactome pathways enriched in both niraparib (green) and rucaparib (purple) datasets following Gene Set Enrichment Analysis. **d** Correlation of effect sizes for genes with expression significantly associated with both niraparib and rucaparib levels per ROI. Drug levels were modelled as continuous variables, derived from mean signal intensities per ROI in MSI images, normalised to heavy-labelled drug-analogue internal standards. Associations were estimated using a two-sided linear mixed model (LMM). The black line represents the LMM regression (*slope* = 0.94, $R^2 = 0.73$) with SE as the grey shaded area. Point size and colour indicate significance of niraparib and rucaparib effects, respectively. **e** Lysosome-related terms identified in a GO Cellular Compartment analysis of genes commonly associated with niraparib and rucaparib. **f** Effect sizes of lysosomal genes significantly associated with niraparib (green) and rucaparib (purple) in PDE samples. **g** Scatter plots depicting expression of lysosomal genes relative to rucaparib concentration, each dot represents an ROI and patients are colour-coded. Black line represents linear regression fit, with SE as the grey shaded area. Pearson correlation coefficients are indicated for each plot; statistical significance was assessed using two-sided Pearson correlation tests. Source data are provided as a Source Data file.

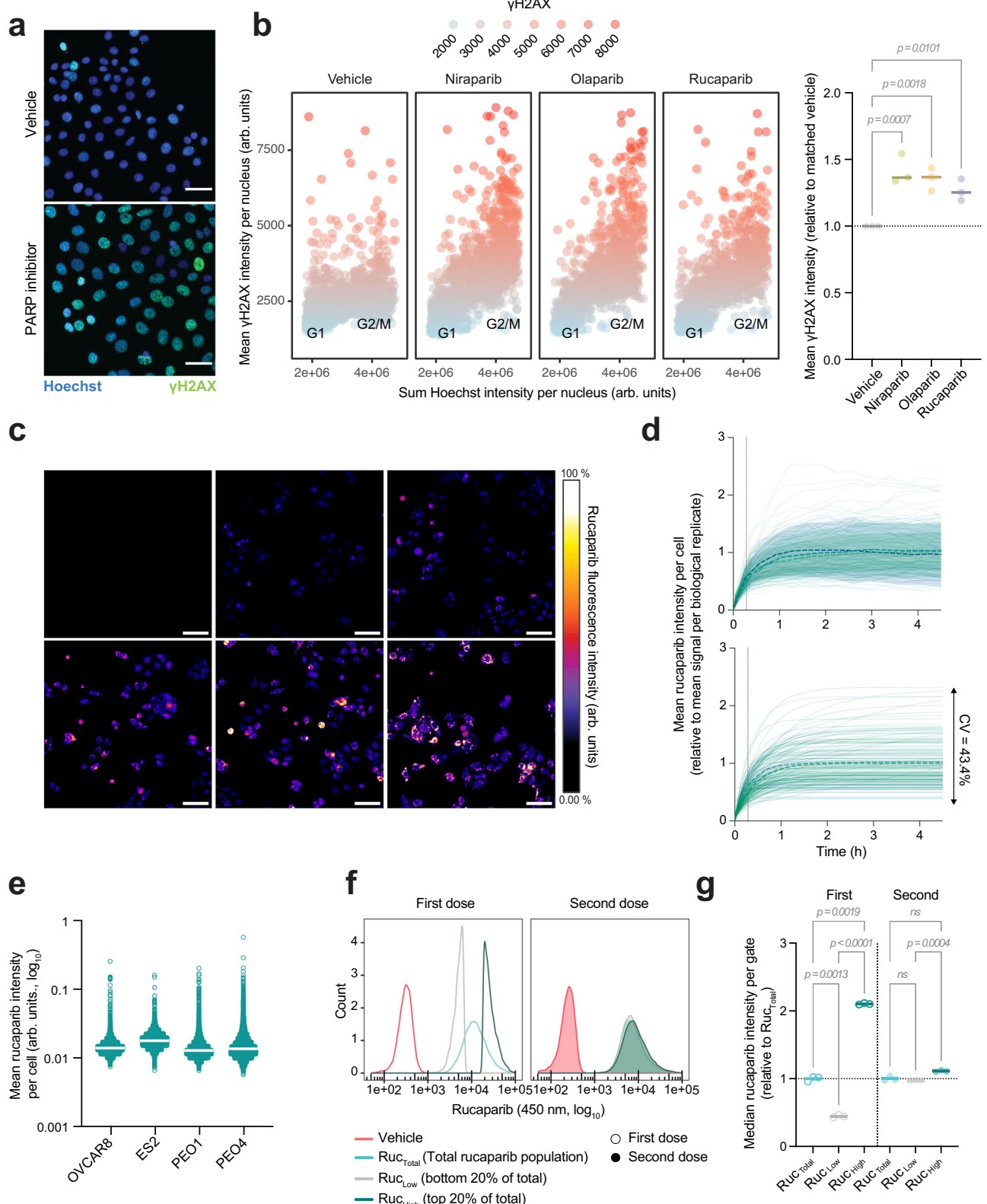

DNA damage response to PARP inhibitors at the single cell level was highly variable in PEO1 and other ovarian cell lines (Fig. 3b, Supplementary Fig. 5c), as demonstrated previously in other cancer cell types[21]. Separating cells based on DNA content revealed a cell cycle-dependent response, in line with the known mechanism of action of PARP inhibitors. As expected, cells in the G2/M-phase demonstrated higher levels of ᵧH2AX compared to vehicle-treated controls.

However, even within this subset, a substantial degree of heterogeneity in drug-induced DNA damage was observed.

**Intracellular rucaparib concentrations are heterogeneous at the single cell level**

Our PDE models suggested that differential drug accumulation is associated with heterogeneity in drug response. To understand

**Fig. 3 | Non-genetic heterogeneity drives variable rucaparib accumulation at the single cell level. a** Representative image of γH2AX levels in PEO1 after 24 h treatment with PARP inhibitors at $IC_{50}$ concentration. Scale bar = 50 μm. **b** Left; Mean γH2AX intensity as a function of sum Hoechst intensity per nucleus upon a 24 h PARP inhibitor treatment. Cell cycle gates were estimated based on Hoechst sum fluorescence intensity per nucleus. A representative biological replicate is shown. Right; biological γH2AX means ($n = 3$) expressed relative to the mean of the vehicle control within each replicate. $P$-values were calculated by a two-sided, repeated-measures one-way ANOVA with Dunnett's multiple comparison test. **c** Rucaparib uptake time course in cells dosed at $IC_{50}$ concentration. Scale = 100 μm. Colour scale depicts drug fluorescence intensity. **d** Rucaparib accumulation across time at the single cell level in PEO1 (top). Dotted lines and shaded areas show mean ± SD of rucaparib uptake, respectively across 50 cells per biological replicate ($n = 3$, colour coded). A one-phase association model was fitted (bottom) to estimate uptake rate and steady-state rucaparib per cell. **e** Heterogeneity in intracellular rucaparib concentration in OvCa cell lines, each dosed at $C_{max}$ of rucaparib (6 μM) for 24 h. Population median rucaparib is showed as solid horizontal lines. Data is representative of $n = 3$ biological replicates. **f** Phenotypic plasticity of differential rucaparib accumulation. $Ruc_{High}$ and $Ruc_{Low}$ PEO1s (top/bottom 20% of rucaparib positive population) were FACS-sorted in triplicates with total drug and vehicle-only controls after a 2 h treatment. Cells were re-cultured in drug-free medium for 21 days and then re-dosed to analyse rucaparib signal distribution, with respect to its original distribution. **g** Relative drug levels for each rucaparib population in (**f**). 'First' denotes initial rucaparib signal. 'Second' refers to re-dosed levels after re-culture. Significance by two-sided Welch one-way ANOVA, followed by Dunnett's. ns: $p > 0.05$. Source data are provided as a Source Data file.

whether this was recapitulated in cell line models, single cell measurements of intracellular drug concentrations were required. Rucaparib has intrinsic fluorescent properties that enable its visualisation by fluorescence microscopy[22], and high-content imaging approaches showed a clear and specific rucaparib signal in PEO1 cells (Supplementary Fig. 6a) that could be visualised as early as 1 minute after addition of the drug, later developing a punctate distribution (Fig. 3c). Quantitation confirmed that intracellular concentrations of rucaparib ($[Rucaparib]_{IC}$) were indeed heterogenous, with a 5-fold difference in signal arising from those cells with the highest versus the lowest drug signal, and a coefficient of variance (CV) of 43.4 ± 9.8% (Fig. 3d). By fitting a one-phase association curve, it was observed that the rate of uptake into cells (K) was inversely correlated to the steady-state $[Rucaparib]_{IC}$ ($r^2 = -0.48 ± 0.12$, Supplementary Fig. 6b). Heterogeneity in $[Rucaparib]_{IC}$ was also observed in other OVCA cell lines, with CVs ranging from 43 ± 16% to 77 ± 12% (Fig. 3e).

To understand whether differential rucaparib uptake in PEO1 cells was due to genetic variability within the population, we treated cells with rucaparib for 2 h (to ensure steady-state $[Rucaparib]_{IC}$ had been achieved, Fig. 3d) and then FACS-sorted the top and bottom 20% of the population, based on rucaparib fluorescence, to obtain $Ruc_{High}$ and $Ruc_{Low}$ populations, respectively (Fig. 3f). These populations were then returned to culture for 1, 2 or 3 weeks before re-incubating with rucaparib and analysing the fluorescence distribution once more, to determine whether the phenotype was stable over several rounds of cell division (Supplementary Fig. 6c, Fig. 3f, g). $Ruc_{High}$ and $Ruc_{Low}$ populations had almost identical fluorescence distributions after 3 weeks in culture. Together, these data demonstrate that the intracellular concentration of rucaparib is highly heterogeneous at the single cell level, and that this heterogeneity is likely to arise from non-genetically encoded molecular differences between single cells within the bulk population.

### Deconvoluting molecular drivers of differential PARP inhibitor accumulation from pharmacological consequences

To determine the molecular drivers of differential rucaparib accumulation, we reasoned that by sorting rucaparib-treated cells at early time points, we would enrich for intrinsic molecular differences between the high- and low-drug populations, while minimising changes driven by differential drug responses (Fig. 4a).

A proteomics analysis of the total population of rucaparib versus vehicle-treated cells after just 1 h of treatment revealed just two significantly changed proteins, suggesting minimal detectable drug response at the protein level at this time point (Fig. 4b). At the same early time point, a large number of proteins were significantly enriched in $Ruc_{High}$ relative to $Ruc_{Low}$ cells, which were inferred to be intrinsic molecular differences between these two populations of cells. Similar expression profiles were also found between $Ruc_{High}$ and $Ruc_{Low}$ cells sorted after 2 and 24 h of treatment, suggesting that the proteins that drive differential rucaparib accumulation are maintained as the cells

respond to rucaparib treatment (Fig. 4b, Supplementary Data 7). Pathway enrichment analysis revealed terms that converged strikingly on lysosomal acidification/autophagy (e.g. LAMP1, LAMP2, CTSL, APP, SCARB2, ATP6V1F) (Fig. 4c, Supplementary Data 8), recapitulating the findings from our PDE samples. The identification of lysosomal signatures at early time points (where drug responses are minimal) in the $Ruc_{High}$ populations suggests that they are an intrinsic difference between these and the $Ruc_{Low}$ population.

$Ruc_{High}$ cells were also enriched in a number of mitotic proteins (e.g. AURKA, aurora kinase; CCNB1 and CCNB2, cyclin B1 and B2; TPX2). Since G2/M phase cells are larger, we looked at the relative cell sizes in our $Ruc_{High}$ and $Ruc_{Low}$ populations. Though $Ruc_{High}$ cells had a 26% increase in forward-scatter signal (FSC-A, which is proportional to cell diameter), this was not comparable to the 5.7-fold difference in median rucaparib signal, and after normalisation to cell volume, there was still an approximately 2.5-fold difference in median rucaparib fluorescence per cell in $Ruc_{High}$ versus $Ruc_{Low}$ populations at early time points, suggesting that increased cell size in the G2/M phase was not a major driver of the increased drug signal in $Ruc_{High}$, but may instead be a correlating feature with cells that have higher lysosomal content[23] (Supplementary Fig. 7a-c).

We had noted previously that rucaparib localisation within cells became punctate over time. Given the lysosomal signatures observed in both patient samples and $Ruc_{High}$ cells - and considering that several drugs are known to localise to the lysosome[24–26]- we hypothesised that rucaparib may accumulate there. Live cell imaging of rucaparib with LysoTracker dye demonstrated that, although initial signal is cytosolic, rucaparib punctae colocalise with lysosomes within 30 min of drug incubation (Fig. 4d, Supplementary Fig. 7d). Cells treated with both rucaparib and LysoTracker displayed a striking correlation between both fluorescent signals, with $Ruc_{High}$ being significantly enriched in LysoTracker signal with respect to $Ruc_{Low}$ (Fig. 4e, Supplementary Fig. 7e, Pearson's $rho = 0.83 ± 0.08$). Image quantification in PEO1 and OVCAR4 cells also revealed that LysoTracker signal was highly correlated with rucaparib levels per cell (Pearson $rho = 0.84 ± 0.05$ and $0.81 ± 0.06$, respectively; Fig. 4f). Transient over-expression of TFEB-GFP, a master regulator of lysosomal biogenesis, significantly increased $[Rucaparib]_{IC}$ demonstrating that lysosomal content changes can drive increased rucaparib accumulation (Supplementary Fig. 7f, g). Conversely, treatment of cells with bafilomycin or chloroquine (CQ) to alkalinise lysosomes decreased $[Rucaparib]_{IC}$ (Fig. 4g, Supplementary Fig. 7h-j). Additionally, the V-ATPase activator EN6[27] triggered an increase in intracellular rucaparib levels (Fig. 4g). Together, these findings demonstrate that heterogeneous $[Rucaparib]_{IC}$ is driven by differential lysosomal content and that this accumulation is pH dependent.

To determine if this pH-dependent lysosomal accumulation is conserved across different cancer types, we compared rucaparib accumulation across a panel of cancer cell lines from diverse origins, including lung (A549), colorectal (RKO), breast (ZR751 and HS578T) and

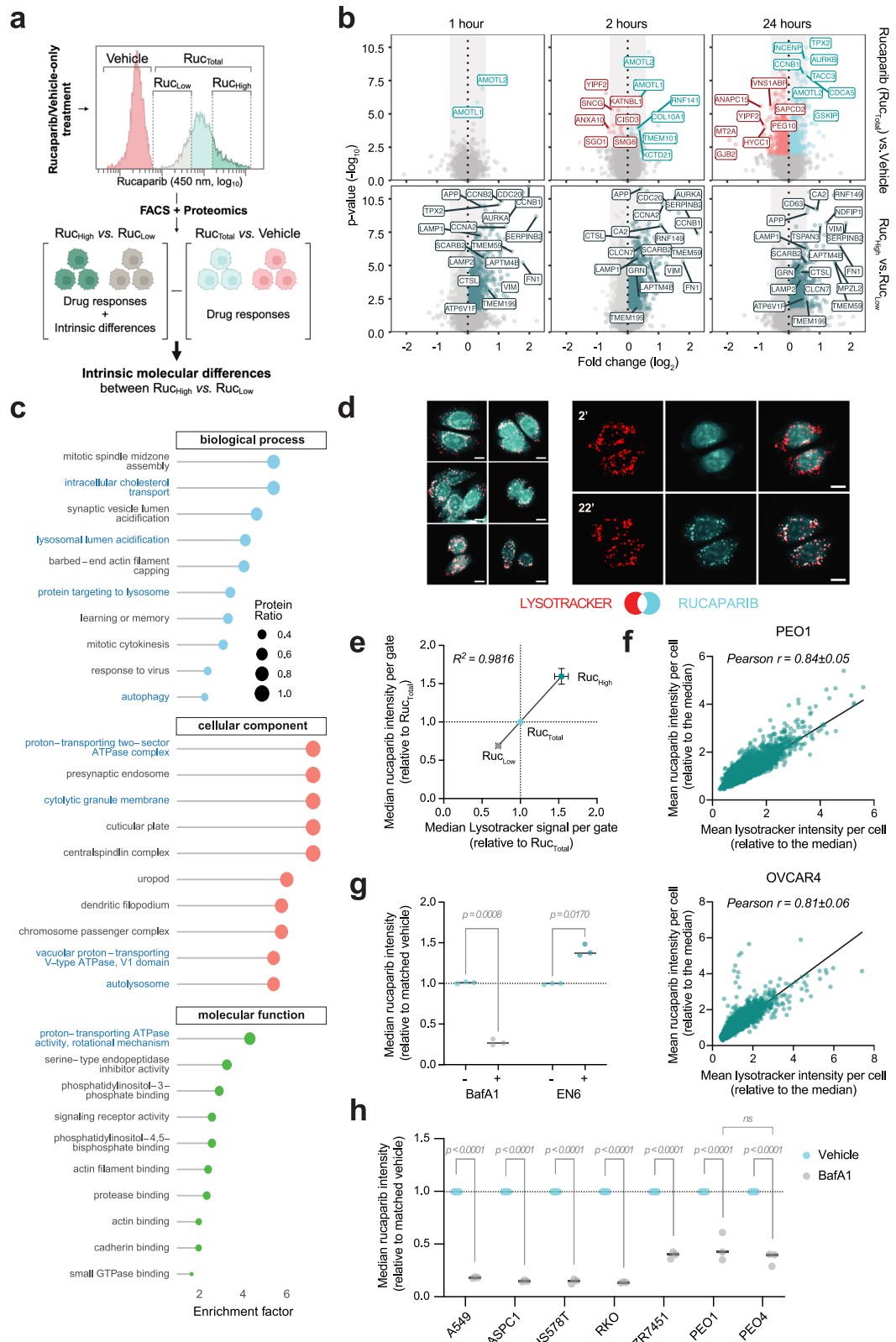

pancreatic (ASPC1) cancers. We also included the isogenic pair PEO1 and PEO4 to rule out an impact of HR status, since PEO4 cells have a mutation that restores the function of the *BRCA2* gene (Fig. 4h). Lysosomal accumulation was quantified by measuring the reduction in rucaparib fluorescence after a 1-h treatment with bafilomycin. We found that all tested cell lines demonstrated significant lysosomal rucaparib accumulation, confirming that this pharmacokinetic property is a universal determinant of rucaparib intracellular accumulation and occurs irrespective of cancer type or genetic background.

## Intracellular rucaparib concentration correlates with DNA damage response in equimolar treated cells

To determine whether differential intracellular concentrations of PARP inhibitors could be a contributing factor to the variable DNA damage

**Fig. 4 | Lysosomal content determines intracellular rucaparib concentration.**
**a** Experimental rationale. Cells were FACS-sorted into $Ruc_{High}$ and $Ruc_{Low}$ (top/bottom 20% of rucaparib positive population) for proteomics analysis. If absent in vehicle-only vs $Ruc_{Total}$ comparison, differential expression between $Ruc_{High}$ and $Ruc_{Low}$ populations represent intrinsic cellular differences pre-existing drug treatment. *Created in BioRender. Ramirez moncayo, C. (2026)* https://BioRender. com/qitpz9o. **b** Volcano plots of differential protein levels in PEO1 cells after 1, 2, or 24 h rucaparib incubation. Data from $n = 4$ biological replicates. Proteins with different abundance across conditions (*q*-value < 0.05) are coloured, with labelling of selected differentially expressed hits. **c** Gene ontology enrichment of up-regulated proteins in $Ruc_{High}$ vs. $Ruc_{Low}$ at 1 h. A *q*-value cut-off of 0.05 was applied to define significance of enrichment. Lysosomal-related term are highlighted in blue. **d** Left: Rucaparib/LysoTracker colocalisation in PEO1 cells (22 min). Right: Distribution changes at 2 and 22 min. Cyan: Rucaparib; Red: LysoTracker. Scale bar = 10 μm.

**e** FACS-based correlation between LysoTracker and rucaparib after 1.5 h dosing of both compounds in PEO1 cells. Median signals per gate (as in **a**) are relative to $Ruc_{Total}$ (mean ± SD across $n = 3$ biological replicates). **f** Imaging-based correlation between mean LysoTracker and rucaparib intensities in PEO1 (top) and OVCAR4 (bottom), after 1 h dosing. Data representative of $n = 3$ biological replicates. Biological mean ± SD Pearson correlation coefficients are shown. **g** FACS-based quantification of median rucaparib signal in PEO1 cell populations. Cells were pre-treated with bafilomycin (BafA1) or EN6 for 1 or 3 h respectively, followed by 1 h with BafA1/EN6 plus rucaparib. Signal normalised to rucaparib-only controls ($n = 3$ biological replicates). **h** FACS-based quantification of lysosomal rucaparib accumulation across cancer cell line models (1 h ± BafA1), estimated by fluorescence fold-change. Cells were treated with rucaparib 10 μM. Data are derived from $n = 3$ biological replicates. For **g** and **h**, significance was assessed by two-way ANOVA and Sidak test. Source data are provided as a Source Data file.

response seen in PEO1 and other OVCA cell lines, measurement of $[Rucaparib]_{IC}$ and γH2AX levels were required within the same cell (Fig. 5a).

PARP inhibitor-induced DSBs occur as a result of PARP trapping and replication fork collapse during S-phase[15]. We therefore synchronised cells in G1 phase with palbociclib[28,29] (Supplementary Fig. 8a, b) prior to releasing them in the presence of rucaparib, to ensure that all cells that had reached G2/M after treatment had had equal opportunity to accumulate DNA damage. We found that indeed, in palbociclib-synchronised cells, there was a positive correlation between $[Rucaparib]_{IC}$ and γH2AX levels, and that this correlation was strongest in cells in G2/M (Fig. 5b, c), in keeping with this population having the largest 'opportunity' to accumulate damage. Furthermore, separation of cells into quartiles based on levels of intracellular rucaparib revealed a statistically significant increase in γH2AX from one quartile to the next (Fig. 5d) with a linear relationship between the median $[Rucaparib]_{IC}$ and median γH2AX fluorescence intensity within each quartile (Fig. 5e). This interaction was also observed in OVCAR4 cells (Supplementary Fig. 8c, d).

To understand the implications of differential rucaparib accumulation on drug response, cells were again sorted into $Ruc_{High}$ or $Ruc_{Low}$ populations. Even after just 2 h of treatment, $Ruc_{High}$ cells subsequently proliferated more slowly and plateaued at a lower cell mass than $Ruc_{Low}$ cells (Fig. 5f, g), further demonstrating that differential uptake at the single cell level impacts rucaparib response. Together, these data demonstrate that despite equimolar dosing under standard cell culture conditions, differences in $[Rucaparib]_{IC}$ contribute to heterogeneity in DNA damage and rucaparib efficacy at the single-cell level. Since lysosomal accumulation is highly correlated to intracellular rucaparib levels, which, in turn, are associated with increased DNA damage, this suggests that lysosomal accumulation of rucaparib increases drug efficacy.

### Lysosomal accumulation increases nuclear bioavailability of weak base PARP inhibitors

Lysosomal localisation of drugs is often referred to as trapping and associated with drug resistance. However, $Ruc_{High}$ cells have both increased lysosomal accumulation and increased drug response. We therefore hypothesised that lysosomes act as a reservoir that maintains $[Rucaparib]_{IC}$ at a higher level and therefore improves bioavailability. Since PARP1 and 2 are nuclear proteins, we quantified nuclear rucaparib per cell, with or without the addition of bafilomycin, CQ, or the V-ATPase activator EN6[27]. Alkalinisation of lysosomal pH with bafilomycin or CQ treatment prevented lysosomal accumulation and substantially decreased nuclear levels of rucaparib in PEO1 cells (Fig. 6a, b, Supplementary Fig. 9a-e). Similar results were seen for OVCAR4 cells treated with bafilomycin (Supplementary Fig. 9h-j). Conversely, increased acidification of lysosomes with EN6 drove increased lysosomal accumulation and elevated nuclear rucaparib

levels in PEO1 cells (Fig. 6a, b, Supplementary Fig. 9f, g). To determine whether lysosomal accumulation consistently sustains greater nuclear rucaparib levels regardless of the extracellular drug concentration, we assessed nuclear drug after manipulating lysosomal pH over a wide range of extracellular rucaparib concentrations (1, 5, and 15 μM). Imaging-based quantification of nuclear rucaparib in PEO1 cells showed that bafilomycin treatment decreases nuclear drug across all tested drug concentrations (Supplementary Fig. 9l). This suggests that lysosomes are likely to act as a reservoir in vivo, even if drug delivery is limited by poor vascularisation. Additionally, upon washout of rucaparib from the extracellular space, rate of fluorescence loss was slower in cells with intact lysosomal function, relative to those treated with bafilomycin or CQ, further supporting the model that the equilibrium of cytosolic and nuclear rucaparib is maintained at a higher concentration by lysosomal accumulation (Fig. 6c, Supplementary Fig. 9k).

Our PDE spatial transcriptomics data suggested an interaction with lysosomes for niraparib as well as rucaparib. Lysosomal accumulation has been documented for a range of drugs[26,30–34], particularly those that are weak bases. These compounds become protonated in the acidic environment of the lysosomal lumen, a process that reduces their lipophilicity (Log*D*) and their ability to diffuse back into the cytosol—ultimately leading to their accumulation within the organelle (Fig. 6d, e). In the case of rucaparib, our data suggest that an equilibrium is maintained between the lysosomal pool of drug and the cytosol/nucleus, with lysosomal accumulation, thereby maintaining higher drug concentrations across the cell. Among the PARP inhibitors clinically used, both rucaparib and niraparib are weak bases, with $pK_a$ values of 9.3 and 10.1 for their most basic moieties, respectively (DrugBank), thus making them susceptible to this mechanism. Conversely, olaparib has a much lower $pK_a$ (−0.9), suggesting it is unlikely to undergo pH-dependent lysosomal trapping (Supplementary Data 2, Fig. 6e). To test this, since neither niraparib nor olaparib is fluorescent, we measured intracellular concentrations of all three PARP inhibitors, with or without bafilomycin pre-treatment, using liquid chromatography-mass spectrometry. Both rucaparib and niraparib intracellular concentrations were significantly decreased in PEO1 and OVCAR4 cells in the presence of bafilomycin, whereas olaparib was unaffected (Fig. 6f, Supplementary Fig. 10a), supporting the hypothesis that only weak base PARP inhibitors accumulate lysosomally.

To confirm that response to weak-basic PARP inhibitors is influenced by differential lysosomal content, we sorted PEO1 cells after incubation with Lysotracker into $Lyso_{High}$ and $Lyso_{Low}$ populations (Supplementary Fig. 10b). While the difference in lysosomal content was partially diminished due to re-equilibration of the population following re-culturing, the $Lyso_{High}$ cells retained a higher capacity for rucaparib accumulation ($Ruc_{High}$) relative to $Lyso_{Low}$ cells ($Ruc_{Low}$, Supplementary Fig. 10c). We then assessed the DNA damage response

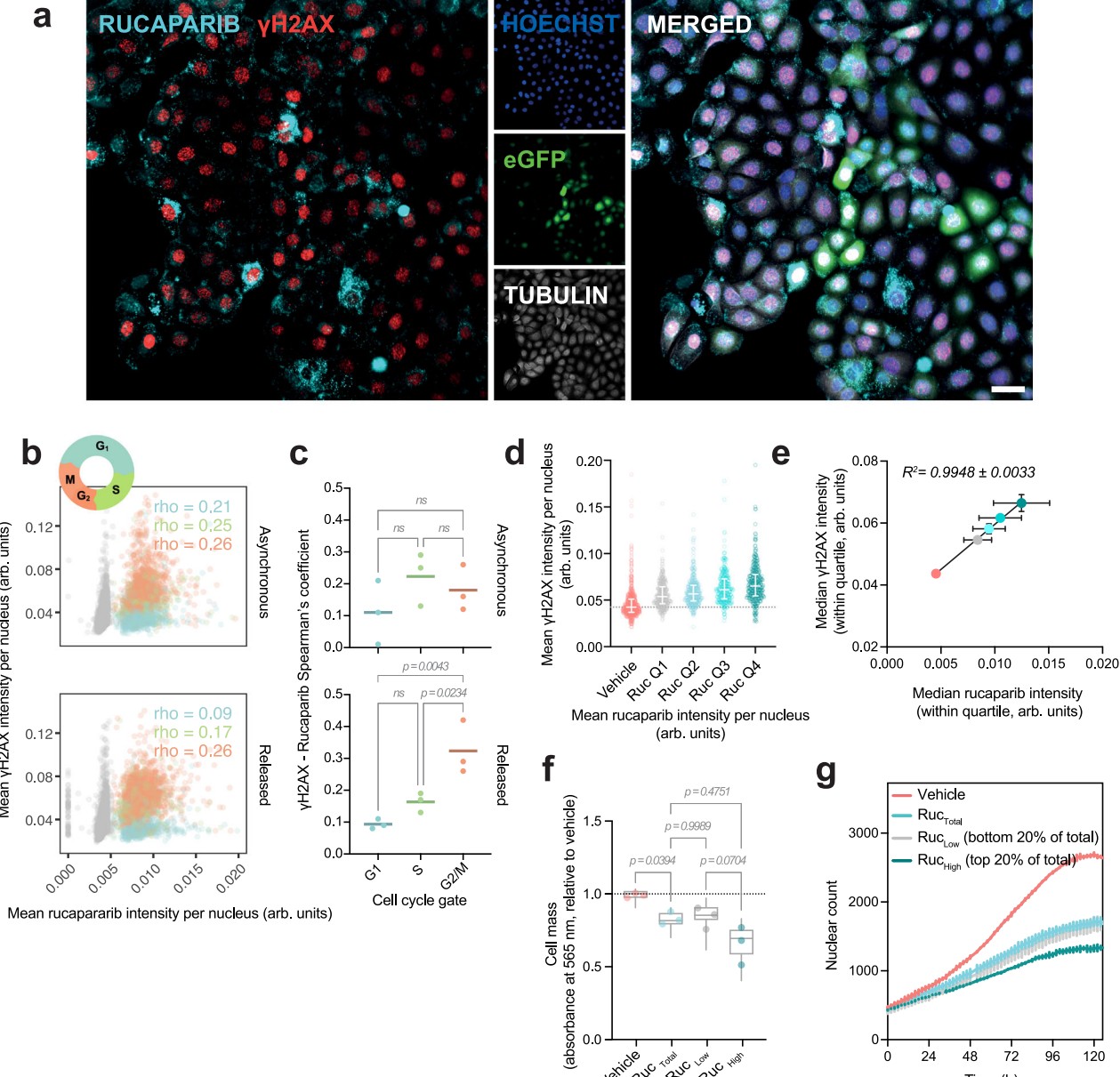

**Fig. 5 | Differences in intracellular rucaparib concentration contribute to heterogeneity in DNA damage levels in equimolar-treated cells. a** Rucaparib and γH2AX imaging in PEO1 cells. Rucaparib (355/480 nm) and eGFP were imaged live. Cells were fixed and immunostained for γH2AX and tubulin, using Hoechst to stain the nuclei. eGFP was imaged a second time to co-register live and fixed-cell images. Scale bar = 50 μm. **b** Palbociclib-treated ('released') and asynchronous PEO1 cells were treated with rucaparib ($IC_{50}$) or vehicle-only for 24 h. Cells were imaged as in (**a**) to correlate rucaparib and γH2AX per nuclei. Representative example of Spearman's correlation across cell cycle gates (blue: G1; green: S; red: G2) is shown, calculated for both asynchronous and palbociclib-released conditions. $p < 0.0001$ for all *rho* values (two-sided Spearman's rank correlation test) except released-G1 condition, with $p = 0.0176$. Greyed points depict vehicle-only population. **c** Cell cycle-resolved Spearman's correlation coefficients between rucaparib and γH2AX per biological replicate ($n = 3$, ns: $p > 0.05$). **d** The G2/M gate of the palbociclib-treated population was subdivided into quartiles based on nuclear rucaparib to compare γH2AX between groups (median ± interquartile range, data representative of $n = 3$ biological replicates). **e** Median γH2AX is linearly related to median intracellular rucaparib within the quartile. Mean ± SD from $n = 3$ biological replicates are plotted. **f** Following a 2 h treatment, cells were FACS-sorted into Ruc$_{High}$ and Ruc$_{Low}$ (top/bottom 20% of the rucaparib positive population). Cells were re-cultured, allowed to proliferate in drug-free medium for 72 h, and stained with SRB to measure cell mass. Boxes show median with min-to-max distribution of technical replicates; dots represent biological means ($n = 3$). **g** Growth curves depicting re-proliferation of the different rucaparib gates ($n = 3$ technical replicates, representative of $n = 3$ biological replicates). Differences in growth rates and plateaus were confirmed in a two-sided extra sum-of-squares F test over the fitted logistic growth model, with $p < 0.0001$. For (**c, f**), significance was tested using one-way ANOVA with Tukey's multiple comparison correction. Source data are provided as a Source Data file.

(by measuring γH2AX foci) in these populations following treatment with either rucaparib or olaparib. Consistent with our hypothesis, the Lyso$_{Low}$ population showed significantly decreased DNA damage levels in response to rucaparib, indicating lower drug bioavailability (Fig. 6g). Crucially, the Lyso$_{High/Low}$ populations showed no difference in their response to olaparib (Fig. 6g), again indicating that lysosomal content only affects the bioavailability and efficacy of weak base PARP inhibitors.

Building on this finding, we sought to test whether lysosomal drug pool acts as a functional reservoir that can sustain drug activity over

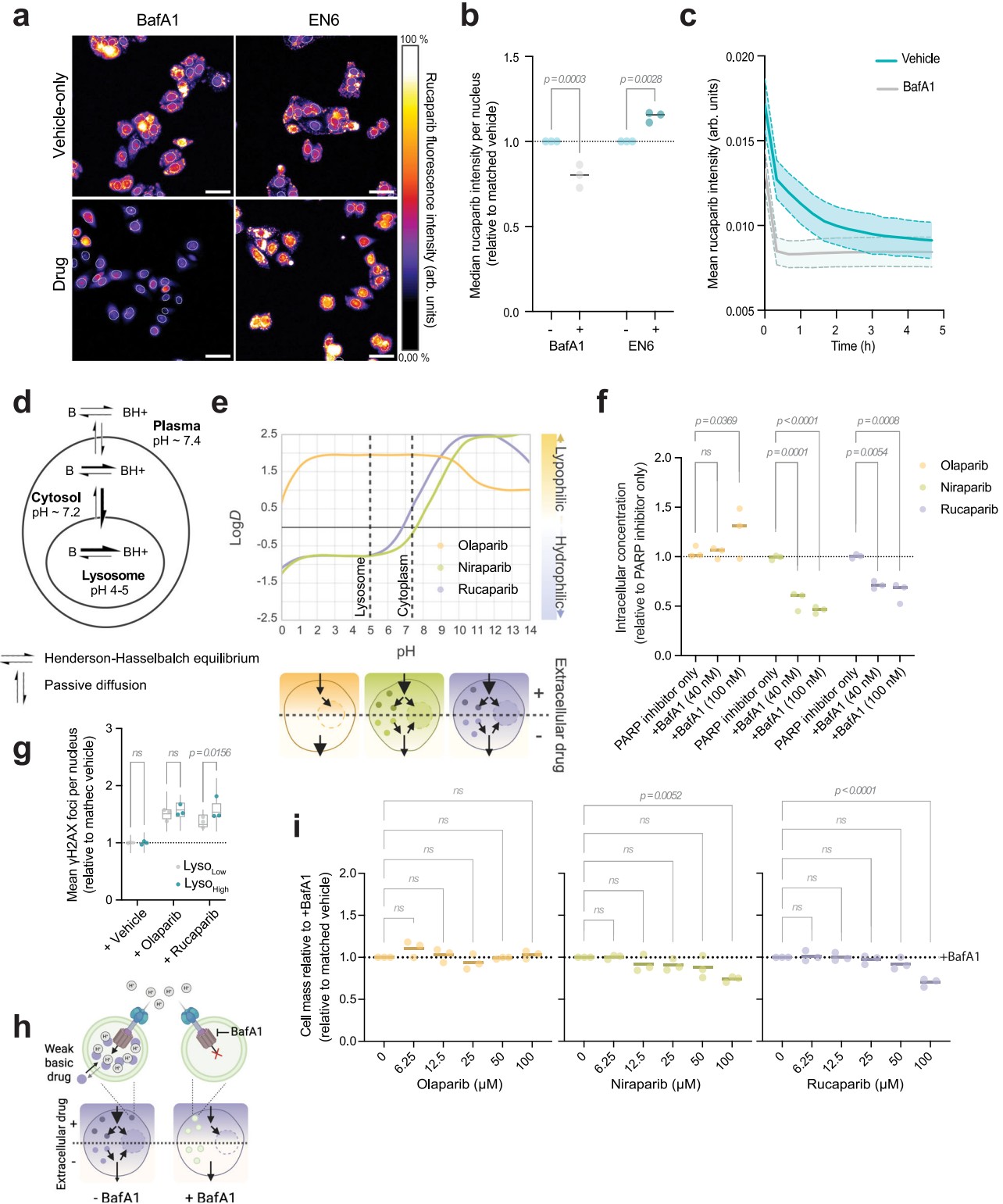

time, even after extracellular drug clearance. If this were the case, then prevention of lysosomal drug accumulation would impact the efficacy of rucaparib and niraparib after short-term incubations but would have no effect on olaparib treatment. To functionally test this, we performed a pulse-washout cytotoxicity assay (Fig. 6h, i). Cells were briefly exposed to PARP inhibitors at different concentrations for 1 h, in the presence or absence of bafilomycin. The drug and bafilomycin were then removed, and cells were allowed to proliferate for 72 h to assess long-term cytotoxicity. As our model predicted, increased

cytotoxicity was observed in cells in which rucaparib and niraparib could accumulate lysosomally, relative to those treated with bafilomycin, which prevents the use of this reservoir. Consistent with all other findings, bafilomycin had no impact on the cytotoxicity of olaparib (Fig. 6i).

Together, these data suggest that as weak bases, rucaparib and niraparib accumulate within the lysosome and that this accumulation increases their bioavailability, whereas olaparib bioavailability is unaffected by lysosomal content or function.

**Fig. 6 | Lysosomal localisation increases nuclear bioavailability of niraparib and rucaparib but not olaparib. a** Representative images of rucaparib (1 h) in PEO1 ± V-ATPase inhibitor bafilomycin (BafA1) or activator EN6 pre-treatment. Nuclei outlined in white. Scale bar = 50 μm. **b** Nuclear rucaparib in PEO1 pre-treated with ± BafA1/EN6 (1 h), normalised to rucaparib-only controls ($n = 3$ biological replicates). **c** Rucaparib loss after washout in cells pre-treated with ± BafA1. Data, representative of 2 biological replicates, shows mean rucaparib per cell ± SD ($n = 6$ technical replicates). **d** pH-dependent partitioning of lipophilic weak bases. Adapted from[30]. **e** Top; Log*D* of olaparib, rucaparib and niraparib across pH. Bottom: schematic illustrating lysosomal accumulation driving increased weak base concentrations and sustained target engagement. *Created in BioRender. Ramirez Moncayo, C. (2026)* https://BioRender.com/5qmjfgl. **f** Intracellular concentration of olaparib, niraparib and rucaparib in PEO1 cells ± BafA1. $n = 3$ biological replicates relative to PARP inhibitor-only. **g** Lyso$_{High}$ and Lyso$_{Low}$ PEO1 cells (top/bottom 20% of LysoTracker-treated population) were FACS-sorted in triplicates. Sorted populations were re-cultured overnight in drug-free medium and treated at $IC_{50}$ levels of rucaparib, olaparib, or vehicle for 24 h in technical triplicates (27 nested observations per condition, $n = 3$ biological replicates). γH2AX foci were quantified relative to matched vehicle per replicate, and analysed using Linear Mixed-Effects Models to account for multi-level nesting. Biological and sorting replicate were included as fixed and random effect, respectively. *P*-values were adjusted via False Discovery Rate (FDR). Boxes show median with min-to-max distribution of technical replicates; dots represent biological means. **h** Effect of V-ATPase inhibition on intracellular concentrations of weak bases. *Created in BioRender. Ramirez Moncayo, C. (2026)* https://BioRender.com/z25hu5c. **i** PEO1 cells were exposed to PARP inhibitors for 1 h ± BafA1 and placed in drug-free medium for 72 h ($n = 3$ biological replicates). Unless otherwise stated, statistical significance was assessed by two-way ANOVA with Sidak's (**b**) or Dunnett's correction (**f**, **i**); all tests were two-sided. ns: $p > 0.05$. Source data are provided as a Source Data file.

## Discussion

The intracellular concentration of a drug is central to determining its target binding and efficacy: too low, and the intended target will not be engaged; too high, and the drug will interact with lower affinity interactors, leading to off-target effects[1]. Despite its importance, drug distribution within tumours and the mechanisms that regulate this are poorly understood. Following equimolar ex vivo dosing in a vasculature-independent, patient-derived system, our novel multi-modal imaging pipeline demonstrated the presence of cell-intrinsic heterogeneity of PARP inhibitor distribution at three levels: between patients, different tumour sites from the same patient, and within individual tumours. Furthermore, regions of high niraparib and rucaparib accumulation in PDEs displayed increased apoptotic signatures, potentially enabling lower-drug accumulating regions to drive therapeutic resistance. Since tumours used here were resected from chemotherapy-naïve patients, our findings suggest that differential accumulation of PARP inhibitors may be a contributing factor to intrinsic resistance to these drugs. This evidences the need to investigate how tumour heterogeneity affects drug distribution as a potential mechanism of intrinsic resistance across cancer therapies and tumour types.

Interestingly, while niraparib and rucaparib exhibited similar relative uptake levels in a given patient, olaparib followed a distinct pattern, suggesting that the mechanisms of accumulation differ between olaparib and the other two drugs. This observation was borne out by the fact that intracellular concentrations of niraparib and rucaparib are decreased by lysosomal alkalinisation, however olaparib is not.

It has been shown that a number of drugs accumulate within the lysosome[26,35], with the suggestion that this occurs because they are weak bases that become protonated within the acidic lysosomal lumen. Lysosomal sequestration is often referred to as 'drug trapping', which limits cytosolic or nuclear target engagement, but this mechanism of resistance remains poorly characterised. Furthermore, the term 'trapping' can be misleading, as weakly basic drugs accumulated in lysosomes may still diffuse back into the cytoplasm under a favourable concentration gradient[30].

Our experiments suggest that the lysosomal pool of rucaparib and niraparib remains active and enhances drug function. This is evidenced by increased DNA damage in Ruc$_{High}$ cells, and decreased nuclear levels of rucaparib when lysosomal accumulation is diminished through bafilomycin or chloroquine pre-treatment. Similarly, in short-term drug loading experiments, prevention of lysosomal accumulation decreased the efficacy of both niraparib and rucaparib, but did not impact olaparib. This fits a model where lysosomes act as a reservoir of rucaparib or niraparib, enabling the cell to maintain an overall higher concentration at the target site, analogous to the accumulation of the tuberculosis drug bedaquiline in lipid droplets[36]. In line with this, we see that rucaparib concentrations are maintained for longer in cells with an intact lysosomal compartment. Our findings highlight the complexity of the role of lysosomes in therapeutic sensitivity and resistance of PARP inhibitors, and other weak base drugs, and warrants further investigation.

Lysosomal biogenesis is regulated by a number of transcription factors, including TFEB, which regulate the expression of CLEAR network genes. The activity of these transcription factors can be linked to several metabolic signalling pathways, such as decreased mTOR signalling or increased AMPK signalling[23,37,38]. Similarly, cGAS/STING signalling, activation of which can be driven by copy number instability (CIN) and micronuclei formation[39], can also drive lysosome biogenesis[40]. Interestingly, CIN and micronuclei formation have recently been demonstrated to display intra-tumoural heterogeneity in ovarian cancer[41], therefore this could also influence the spatial variance in lysosomal content. In cell lines, it is clear that the variability in lysosomal content is driven by non-genetic factors, and though we are unable to conclude this with the data available from our PDE samples, it would be of significant interest in future studies to understand whether any of these factors drive the substantial heterogeneity in lysosomal content observed within HGSOC tumours.

We chose to study those PARP inhibitors indicated for clinical use in ovarian cancer, however several others are licenced or in clinical trials for other cancer types (Supplementary Table 2). Several of these are weak bases with substantial changes in their Log*D* between cytosolic and lysosomal pH, and we would therefore hypothesise that veliparib, saruparib (AZD5305, PARP1 specific) and palacaparib (AZD9574, brain penetrant, PARP1 specific) also become accumulated in the lysosome, while others, including olaparib (as demonstrated experimentally), are likely to accumulate through a lysosome-independent mechanism. This difference could suggest that if poor pharmacodynamics drives lack of response to one PARP inhibitor within a patient's tumour, treatment with another may still be effective. Our multi-modal imaging study also opens the possibility of personalised therapy, where biopsy samples can be used to assess drug accumulation and to determine which drug is likely to be most effective for a patient.

It is important, however, to acknowledge the limitations of our proof-of-concept study, such that they can be addressed in subsequent work. Firstly, we applied our multi-modal pipeline to explants derived from three patients, and while the approach taken enabled us to obtain mechanistic insight into the heterogeneous drug distribution observed, this study cannot have captured the full extent of patient-to-patient variability known to exist clinically. In the future, studies which include larger sample numbers are warranted, ideally including those from patients pre- and post-relapse to assess whether changes in lysosomal content could contribute towards PARP inhibitor resistance. As publicly available datasets linking transcriptional profiles of PARP inhibitor-treated tumours to patient outcomes become available, it would also be pertinent to explore these for links to lysosomal content, to further strengthen clinical relevance of this work. Finally, as

mentioned previously, our work used an ex vivo-dosed explant model to explore cell-intrinsic differences in PARP inhibitor accumulation. Clinically, PARP inhibitors are dosed orally and will reach the tumour via the vasculature, whose often disorganised structure is likely to further increase the heterogeneity of distribution of these drugs across the tumour. At the cellular level, we demonstrated that lysosomal content determines nuclear rucaparib availability, independently of the extracellular concentration (Supplementary Fig. 9 l). However, further investigation would be required to determine whether high lysosomal sequestration in perivascular cells or the tumour periphery could drive extracellular tissue-level gradients by limiting drug penetration into deeper tumour regions. While such gradients are a known challenge for other agents (such as certain formulations of doxorubicin[42,43]), the high clinical $C_{min}$ and $C_{max}$ concentrations of PARP inhibitors[44,45] may help maintain a more uniform steady-state distribution over time. Since there is a therapy-free window prior to surgery, it is very difficult to assess heterogeneity in drug distribution after patients are dosed orally. Future work could use in vivo models to address the extent to which vascular PARP inhibitor delivery could impact drug distribution, although our work suggests that even when delivery is limited, the lysosome is still likely to act as a drug reservoir.

# Methods

## Spatially resolved approaches using patient-derived tumour explants

Human HGSOC tumour samples were obtained through intraoperative tumour mapping[46,47] from chemo-naive patients undergoing maximal effort upfront cytoreductive surgery at Hammersmith Hospital, London, UK, a tertiary gynaecological cancer centre, certified by European Society of Gynaecological Oncology (ESGO) as centre of excellence for ovarian cancer surgery[48,49]. The human samples for this research project were banked by the Imperial College Healthcare Tissue Bank (ICHTB). ICHTB is supported by the National Institute for Health Research (NIHR) Biomedical Research Centre based at Imperial College Healthcare NHS Trust and Imperial College London. ICHTB is approved by Wales REC3 to release human material for research (22/WA/2836). All patients gave written consent. All procedures involving human participants were done in accordance with the ethical standards of the institutional and/or national research committee and with the principles of the 1964 Declaration of Helsinki and its later amendments or comparable ethical standards.

**Tissue slicing and culture.** Prior to processing, samples were kept on ice. PDEs were obtained using a vibrating blade microtome (LeicaVT1200 S, Leica Biosystems, Nussloch, Germany) to cut slices of 400 to 600 μm in thickness. 5% agarose in PBS was used to embed tumour samples prior to slicing, which was performed in PBS supplemented with 100 U/mL penicillin, 100 μg/mL streptomycin (P/S). Resulting PDEs were placed onto inserts (0.4 μm porosity, PICM01250, Merck Millipore, Burlington, MA, USA) within 6-well plates and just submerged for acclimatisation overnight in complete medium (RPMI 1640, 20% FCS (foetal calf serum), 2 mM sodium pyruvate, 100 U/mL penicillin, 100 μg/mL streptomycin, 2.5 μg/mL insulin) at 37 °C, 5% $CO_2$.

**Ex vivo dosing with PARP inhibitors.** Explants were treated with niraparib, olaparib and rucaparib at 20 μM in complete medium or with 0.1% DMSO as vehicle. PARP inhibitors have a $C_{max}$ (source: DrugBank) in the micromolar range (12.4–17.5 μM, 6 μM and 2.5 μM for olaparib, rucaparib and niraparib, respectively), therefore the concentrations used for explants are in line with those observed in patients. PDEs were incubated with drug (37 °C, 5% $CO_2$) for between 2 and 48 h. Typically, 24 h treatment was used to ensure homogeneous drug distribution throughout the tissue slice. Post-incubation, ice-cold drug-free medium was used to rapidly wash PDEs before rinsing with ice-cold PBS

(Supplementary Table 1). Excess liquid was carefully blotted before snap-freezing in liquid nitrogen. Samples were stored at −80 °C.

**Embedding and cryosectioning.** To aid cryosection, frozen PDEs were embedded in a hydrogel (7.5% hydroxypropyl methylcellulose, 2.5% PVP)[50]. Polymer blocks were snap-frozen using dry ice-chilled isopropyl alcohol. 10–12 μm serial sections were cut using a cryostat (CM3050s, Leica Biosystems) set to −17 °C, and thaw-mounted onto slides (12-550-15, Thermo Fisher Scientific, Waltham, MA, USA) prior to storage at −80 °C.

**Histochemistry and immunohistochemistry.** Acetone was used to fix sections (−20 °C). H&E staining was undertaken using the commercial kit ab245880 (ab210904, Abcam, Cambridge, UK) according to manufacturer's instructions, before mounting with DPX. For IHC, endogenous peroxidase activity was quenched using 3% hydrogen peroxide. Sections were then incubated with protein blocking buffer (Abcam) for 1 h at room temperature, followed by overnight incubation at 4 °C with primary antibodies diluted in either SignalStain Antibody Diluent (8112, CST, Danvers, MA, USA) or the same blocking buffer. See Supplementary Table 2 for primary antibody details. Slices were then incubated with secondary antibodies for 1 h (anti-rabbit IgG conjugated with polymeric horseradish peroxidase linker (Leica Bond Polymer Refine Detection, DS9800, Leica Biosystems) for γH2AX and cleaved caspase-3 primaries; biotinylated goat anti-rabbit IgG (HRP/DAB detection IHC Kit, ab64261, Abcam) for the rest). DAB was used as chromogen and sections were haematoxylin counterstained and mounted with DPX. Slides were imaged using a slide scanner (ZEISS Axioscan Z1, Carl Zeiss, Oberkochen, Germany) and a DM4b/DM6000 upright setup (Leica Microsystems, Wetzlar, Germany) and processed with Fiji (v2.16, National Institutes of Health, Bethesda, MD, USA) and QuPath[51,52].

**Mass spectrometry imaging (MSI).** Tissue sections were thawed under low vacuum for 20–30 min. For method development and standard curve calculation, 0.15 μL of compound of interest at known concentrations (diluted in 50% MeOH, 1% DMSO as vehicle) was spotted onto desiccated tissue sections, before further processing. Two MSI platforms were used as described below. Initial set up of our multimodal imaging pipeline (Supplementary Fig. 1e, f) used Desorption Electrospray Ionisation (DESI-MSI) which has a lower spatial resolution but improved speed of acquisition. All MSI completed as part of the main study (Fig. 1, Supplementary Fig. 2) used the MALDI-MSI platform to maximise spatial resolution.

**Matrix-assisted laser desorption/ionization (MALDI) MSI.** Matrix (2,5-Dihydroxybenzoic acid, Sigma-Aldrich, St. Louis, MO, USA, Supplementary Table 1) was applied to the tissue section, which was held by a rotating target (350 rpm) at 5 μL/min (nitrogen flow rate of 5 L/min) for 30 min using a pneumatic sprayer (SMALDIprep, TransMIT GmbH, Giessen, Germany). Heavy-isotope labelled versions of each of the PARP inhibitors ([$^{13}C_6$]-Niraparib, [$^2H_8$]-Olaparib and [$^{13}C,^2H_3$]-Rucaparib; Aslachim, Mont-Saint-Aignan, France) were spiked into the matrix solution at 3 μg/mL as internal standards (IS) as described elsewhere[53]. Positive ion mode was used to acquire spectral data, typically between 250 and 1000 m/z, using an AP-SMALDI5-AF ion source (TransMIT GmbH) coupled to a Q-Exactive Plus mass spectrometer (Thermo Fisher Scientific). Source conditions: capillary temp 350–400 °C; voltage, 4 kV. Mass spectrometric parameters: inject time, 250 ms; 70,000 mass resolution; 3.8 scans/s. Scanning was performed in 2D-line mode with a nominal pixel size set to 20 μm, and laser attenuator set to 34 arbitrary units.

After acquisition, cold ethanol was used to remove matrix to enable H&E staining of the tissue section. Raw spectral data were converted to imzML format (RAW2IMZML converter v1.8R3, TransMIT

GmbH). MSiReader[54] (v1.02, North Carolina State University, Raleigh, NC, USA) was used for data analysis. Normalisation to the appropriate heavy labelled IS was performed. A tolerance window of 5-10 parts per million (ppm) was used to plot features of interest.

**Desorption electrospray ionization (DESI) MSI.** For DESI imaging a source comprising a 2D sample holder moving stage and custom-built inlet capillary (490 °C) (Prosolia, Indianapolis, IN, USA), coupled to a XEVO G2-XS QToF (Waters Corporation, Milford, MA, USA) was used. DESI parameters were as follows: spray voltage, 4.5 kV; solvent, methanol/water, 95:5; flow rate, 2 μL/min; nebulizing gas, nitrogen; gas pressure, 5 bar; sprayer incidence angle, 75°; collection angle, 10°; sprayer-to-inlet distance, 1 mm; sprayer-to-sample distance, 1 mm. Mass spectrometric parameters: source temperature, 150 °C; source offset −80 V; mass resolution 20,000; 1 scans/s. A nominal pixel size of 75 μm was used. Spectral data were acquired in both positive and negative ion modes between m/z 50 and 1000. Images were generated using HDImaging v1.4 and MassLynx v4.1 software (Waters Corporation), applying total ion count (TIC) normalisation, with m/z tolerance value of 20 ppm.

**GeoMx DSP spatial transcriptomics.** Adjacent tissue sections to those used for MSI were processed as recommended by NanoString (Bruker Spatial Biology, Bothell, WA, USA). Briefly, samples were thawed and fixed overnight in 10% neutral buffered formalin (NBF), and washed in PBS before baking for 30 min at 60 °C. They were then dehydrated with successive washes (5 min) in increasing ethanol concentrations. Antigen retrieval was performed at 100 °C for 15 min in Tris-EDTA. Proteinase K was used at 1 μg/mL in PBS to expose RNA targets (37 °C, 15 min). Samples were incubated again in 10% NBF for 5 min and washed twice in NBF stop buffer (Supplementary Table 1). In situ hybridization using NanoString's Whole Transcriptome Atlas probe mix (overnight, 37 °C) was followed by two 25 min incubations in stringent wash (Supplementary Table 1, 37 °C) to remove off-target probes. Samples were blocked with buffer W (30 min, NanoString). Finally, hybridised tissue samples were stained with the nucleic acid dye SYTO-13 and the fluorophore-conjugated, primary antibodies (1 h at room temperature) detailed in Supplementary Table 3. Hybridised and stained samples were scanned on the GeoMx DSP platform (v3.1.2.12), and regions of interest (ROI) selected, based on overlaid MALDI-MSI images of PARP inhibitor distribution in adjacent tissue sections and morphology marker staining. Typically, a minimum of 80 nuclei were selected for each ROI to ensure sufficient sequencing coverage. PCR amplification/library preparations were carried out according to manufacturer's instructions. Pooled libraries were purified using AMPure XP beads (Beckman Coulter, Brea, CA, USA), and final library quality and quantity assessed using the Agilent 2100 High Sensitivity DNA (Agilent Technologies, Santa Clara, CA, USA) and Qubit High-Sensitivity DNA (Thermo Fisher Scientific) assays.

Sequencing was performed using the Illumina NextSeq2000 platform (Illumina, San Diego, CA, USA) (Paired End 27 bp + dual 8 bp indexing). Sequenced data were analysed using the GeoMx NGS pipeline in GeoMx DSP platform v3.1.2.12. Quality control thresholds: 100,000 raw reads and 50% sequencing saturation. Upper quartile (Q3) normalisation was applied across all targets to account for differing ROI size and cellularity which impact transcript abundance. ROIs with 5% or more targets above the limit of quantification (LOQ, defined as negative probe geomean x negative probes geometric standard deviation) were further processed. Only those targets above the LOQ in at least 10% of ROIs were included in the analysis. Genes that were differentially expressed in high-drug relative to low-drug ROIs were identified using a linear mixed model (LMM), using 'Patient ID' and 'PDE slice ID' as correction variables. Benjamini and Hochberg False Discovery Rate multiple test correction was applied, with a significance

threshold of $q < 0.05$. Gene Set Enrichment Analysis[55] was carried out using Reactome[56].

**Modelling spatial transcriptomics data using drug concentration as a continuous variable.** Following quality control and Q3 normalisation, we used the R package *lme4* for linear mixed model analysis of relationship between local drug concentration and gene expression. To model drug concentration (DIC) as a continuous variable (rather than binary high versus low), drug levels within each ROI were quantified from MSI images, calculating mean ion intensity per ROI, relative to heavy-labelled drug-analogue internal standards. To address the hierarchical structure of the data, we employed a linear mixed model with patient and PDE slice as nested random effects, accounting for both inter-patient variability and intra-tumour variability across different slices. The nesting structure recognises the likelihood that slices from the same patient will exhibit more similarity than slices from different patients. The model is represented as:

$$Gene\_Expression \sim DIC + (1|PatientID) + (1|PatientID : sliceID)$$

This model was applied to each drug:gene combination separately, and allowed random intercepts at both patient and slice levels, capturing the inherent variability at each level of the hierarchy. To assess the significance of drug effects on gene expression, the $p$-value was calculated by comparing the full model (including DIC as a predictor) with the null model (excluding DIC), using the R anova() function:

$$Full\ Model : Gene\_Expression \sim DIC + (1|PatientID) + (1|PatientID : sliceID)$$

$$Null\ Model : Gene\_Expression \sim (1|PatientID) + (1|PatientID : sliceID)$$

Functional enrichment analysis was performed using Gene Ontology (GO) terms and Reactome pathways. GO term enrichment analysis used the *gseGO* function in the *clusterProfiler* package. Pathway enrichment was analysed using the *gsePathway* function of the *ReactomePA* package.

## General cell-based methods

**Cell line handling.** PEO1 and 4 were a gift from Simon Langdon. ES2, OVCAR8, OVCAR4 and Kuramochi lines were all obtained from ATCC (Manassas, VA, USA). OVCAR4 EGFP-PARP1-mCherryFP cell line was generated previously[57]. All cell lines were recently authenticated using STR profiling. Cell lines were maintained in RPMI 1640 supplemented with 2 mM Glutamine, 100 U/mL penicillin, 100 μg/mL streptomycin, 10% FCS at 37 °C, 5% $CO_2$ and passaged three times per week using TrypLE express.

**Drug dosing.** Supplementary Table 4 details drug used throughout the study. Stock aliquots in DMSO were stored at −80 °C. For use, drug stocks were diluted in medium, with a maximum vehicle concentration of 0.1%. Chloroquine stocks were made fresh immediately before use.

**Sulforhodamine-B cell mass accumulation assay.** For the estimation of drug $IC_{50}$ values, cells were plated in 96-well plates (3596, Corning, Corning, NY, USA) at 4000-6000 cells/well, (cell line-dependent) in complete medium and left to adhere overnight. Cells were dosed as indicated and after 72 h, plates were fixed with 10% trichloroacetic acid (1 h, 4 °C) and stained with 0.4% Sulforhodamine-B (SRB, 30 min room temperature, Supplementary Table 1). Plates were washed (1% acetic acid) and air-dried before solubilising the stain (200 μL, 10 mM Tris-Base) and reading optical density at 565 nm using SoftMax Pro (v7.3).

**Growth curves.** Cells were seeded on a clear bottom, white 96-well plate (3610, Corning) in media, ± 1:1000 dilution of IncuCyte NucLight Rapid Red Reagent (4717, Sartorius, Göttingen, Germany) to label nuclei, and cultured at 37 °C, 5% $CO_2$ while imaging every 2 h in an Incucyte S3 (v2022B rev3, Sartorius). Confluency and/or nuclear number were plotted.

**Immunofluorescence and high content analysis microscopy.** Cells were seeded at approximately 40% confluency on 96-well plates (clear bottom, black, 6055300 Phenoplates, PerkinElmer, Waltham, MA, USA) and the following day, treated with drug (concentrations and timings indicated in text or legends), before fixing with 4% paraformaldehyde (20 min at room temperature). Blocking and permeabilization were combined using 2% bovine serum albumin (BSA), 0.5% Triton-X in PBS for 30 min. Primary antibodies in Supplementary Table 5 were incubated overnight at 4 °C at the specified concentration in 2% BSA. Fluorophore-conjugated secondary antibodies (Supplementary Table 6) were incubated in the dark, typically at a 1:1000 dilution, for 1 h at room temperature. Hoechst 33342 DNA staining was performed prior to imaging (1 μg/mL in PBS, 15 min). Image acquisition and analysis were carried out using the Operetta CLS High Content Analysis System (Revvity, Waltham, MA, USA), using the 20x air NA 0.8 objective unless otherwise stated. Five images per well of a single focal plane were acquired using a random pattern while excluding the well borders. Focus was manually set. Harmony Software (v4.9, Revvity) was used for flatfield correction, and nuclear segmentation was based on Hoechst staining. Measurements such as Hoechst sum intensity per nucleus (cell cycle gating), as well as γH2AX fluorescence intensity, were recorded, and downstream analysis used custom scripts in RStudio (v2025.05.0 + 496). Further additional image processing used Fiji (v2.16)[51].

**High-content microscopy of rucaparib.** Rucaparib was imaged in live cells that had been washed in FBS-containing medium post treatment immediately before imaging (using the Hoechst channel in Supplementary Table 7), with the Operetta CLS High Content Analysis System (set at 37 °C, 5% $CO_2$), using the 20x air NA 0.8 objective unless otherwise stated. In some experiments, cells were pre-incubated at 37 °C with LysoTracker Deep Red (L12492, Thermo Fisher Scientific) (50 nM, 30 min, imaged using the Alexa Fluor 647 channel in Supplementary Table 7) and/or CellTracker Green (C7025, Thermo Fisher Scientific) (10 μM, 15 min, imaged using the Alexa Fluor 488 channel in Supplementary Table 7) and pre-treated with CQ, bafilomycin or EN6 (typically at 25 μM, 50–100 nM and 50 μM respectively) for 1 h before rucaparib was added for a further 1 h. When needed, samples were fixed and immunostained following live imaging (as above). Custom Fiji macros were developed to align the resulting images based on the identification of common features from both image sets. Typically, eGFP-overexpressing (see lentiviral transduction below) or CellTracker stained cells were used to enable the initial live and then fixed 488 signal co-registration. Pearson correlation coefficients between the superimposed 488 channels were calculated pre- and post-alignment to quality-control the registration process. Should the post-alignment coefficient be lower than the original, the image set was flagged and excluded from analysis. CellProfiler (v4.2.8)[58] was used to analyse aligned images. α-tubulin and Hoechst staining were used to segment individual cells and nuclei, respectively. Cytoplasmic segmentation was achieved by subtracting nuclear space from individual segmented cells. Rucaparib fluorescence intensity was quantified for each segment (cell, cytoplasm, nucleus), while γH2AX and Hoechst fluorescence intensities were recorded for each nucleus. Downstream analysis was carried out using custom scripts in RStudio.

Dynamics of rucaparib uptake were captured using eGFP-overexpressing cells (37 °C and 5% $CO_2$), imaging at 20-minute intervals over 24 h. Image acquisition was performed as described before.

Subsequently, cells were segmented based on their eGFP signal using the Cellpose (v2.0)[59] model 'cyto', employing flow, cell probability, and stitch thresholds of 0.4, −2, and 0, respectively. The resulting cell masks were then imported into Fiji (v2.16)[51] using the BioImaging and Optics Platform (BIOP) plugin to quantify rucaparib fluorescence intensity per cell over time. To characterize rucaparib uptake kinetics, a one-phase exponential association or asymptotic growth non-linear model was fitted over the fluorescence intensity values per cell and time point, using GraphPad Prism (v 10.4.1, GraphPad Software, San Diego, CA, USA).

Rucaparib washout experiments were performed similarly to uptake experiments described above, by pre-staining cells with Cell-Tracker Green and subsequently pretreating with CQ or bafilomycin A1 for 1 h before rucaparib was added for a further 1 h. PBS or DMSO were used as vehicle controls for CQ and bafilomycin, respectively. At time 0, drug-containing medium was removed and replaced with fresh, drug-free complete RPMI media. Cell tracker green and rucaparib were imaged at 20-minute intervals for 5 h using the Operetta. The resulting images were analysed in CellProfiler (v4.2.8) to measure average rucaparib per cell, using CellTracker green for cell segmentation. Mean rucaparib fluorescence intensity per cell was then plotted for each of 6 replicate wells per condition, and a one-phase decay model was fitted in GraphPad Prism (v10.4.1) to calculate half-life per condition.

**FACS sorting.** Cells were treated with LysoTracker at 50 nM for 1 h, or rucaparib at $IC_{50}$/vehicle for 1, 2 or 24 h before washing with PBS, and harvesting using TrypLE Express. Cells were resuspended at 5-10 million cells/mL in ice-cold PBS supplemented with 2% FCS. For experiments requiring re-culture of cells post sorting, phenol red-free RPMI with 2% FCS, 100 U/mL penicillin, 100 μg/mL was used instead. Cell suspensions were filtered through a 35 μm cell strainer (352235, Falcon, Corning, NY, USA) and kept on ice prior to sorting on a FACSAria™ Fusion cell sorter (BD Biosciences, San Jose, CA, USA). Rucaparib was excited by the 355 nm laser and emission was collected using a 450/50 nm bandpass filter. BD FACS Diva v9.4 was used for subsequent analyses, and the top and bottom 20% of rucaparib positive cells were sorted and collected into 50% FCS in PBS or phenol red-free RPMI. Samples and collection tubes were maintained at 4 °C during sorting and kept on ice prior to downstream processing. For re-culturing, cells were centrifuged at 400 × $g$ and resuspended in complete media. Data analysis was performed with FlowJo (v10.10).

**Molecular biology and biochemistry**
**Lentiviral transduction.** HEK293-T cells (grown in DMEM, 100 U/mL penicillin, 100 μg/mL streptomycin, 10% FCS at 37 °C, 5% $CO_2$), were used for viral particle generation. Cells were transfected using FuGene-6 (Promega Corporation, Madison, WI, USA) according to manufacturer's instructions. Briefly, transfection mix was made by combining serum-free DMEM with FuGene, incubating 5 min (room temperature) before adding relevant DNA constructs and further incubating at room temperature (25 min). Transfection mix was added dropwise to plates containing HEK293T-cells. After 6 h, medium was refreshed with complete DMEM and once more the day after transfection. At 48 h post transfection, viral particles were harvested by filtering medium using a 0.45 μm filter (E4780-1456, Starlab, Hamburg, Germany) prior to adding polybrene (final concentration 4 μg/mL). Virus-containing media were stored at −80 °C if not used immediately. For viral transduction, cells of interest were seeded in complete RPMI (~50% confluency), and the following day, were incubated for 6 h with virus-containing medium (37 °C, 5% $CO_2$). Post-infection, cells were allowed to recover for 16 h before puromycin selection (1 μg/mL).

**Transient transfection.** Cells were seeded in 6-well plates 24 h prior to transfection. Cells were transfected with FuGene, using a 1:3 ratio of DNA to transfection reagent, according to manufacturer's instructions.

Briefly, FuGene-6 (9 μL/well) was added dropwise to serum free medium (138 μL/well) and allowed to incubate for 5 min. 3 μg of plasmid (Supplementary Table 8) was added to the mix and further incubated for 25 min before adding dropwise to the well. Media were replaced after 5–7 h with fresh complete RPMI, and cells were used for plating after an overnight recovery.

**Protein extraction and western blotting.** Protein was extracted in Laemmli buffer (Supplementary Table 1) with 100 units/mL benzonase and quantified in a Pierce bicinchoninic acid (BCA) assay (23227, Thermo Fisher Scientific). Typically, 20–40 μg protein were denatured for 5 min at 95 °C and run per lane in a 4–15% mini-Protean precast protein gels (456-1084, Bio-Rad Laboratories, Hercules, CA, USA) in Tris-glycine running buffer (Supplementary Table 1). Protein was transferred to a nitrocellulose membrane (162-0112, Bio-Rad) using the Trans-Blot Turbo Transfer System (Bio-Rad, 15 V, 1.5 A, 25 min) and Bio-Rad protein transfer buffer, stained with ponceau S solution and blocked for 1 h in 5% milk in TBS-T (0.05% Tween-20 in Tris-Buffered Saline). Membranes were then probed with primary antibody (TFEB, D2O7D and β-actin, AC74), diluted 1000 times in 5% BSA in TBS-T with 0.05% sodium azide, at 4 °C overnight, followed by 3 consecutive washes in TBS-T and a 1 h incubation with secondary antibody in 5% milk in TBS-T.

**Proteomic Liquid chromatography mass spectrometry—sample preparation and analysis.** Experiments were performed in 4 independent biological replicates. PEO1 cells were treated in duplicates, with rucaparib at $IC_{50}$ concentration or vehicle only, for 1, 2 or 24 h. Treated cells were FACS-sorted as described before into $Ruc_{High}$ and $Ruc_{Low}$ populations (approximately 1 million cells per replicate), and washed twice with ice-cold PBS before lysis in surfactant cocktail[60] (2% SDS, 1% SDC, and 2% IGEPAL CA-630), with 100 units/mL benzonase added prior to lysis. Protein was measured by BCA assay and concentration adjusted to 2 g/L. 40 μg of lysate was reduced and alkylated by addition of chloroacetamide and TCEP (tris(2-carboxyethyl)phosphine)) to a final concentration of 20 mM and 10 mM, respectively. Samples were precipitated using 200 μg of beads, washed with 100 μL 80% ethanol three times before digesting overnight with 40 μL of 50 mM ammonium bicarbonate buffer containing 20 ng/μL trypsin (Pierce™ P/N 90059) and 10 ng/μL LysC (WAKO) with shaking (1700 rpm). An Ultimate 3000 RSLC nano liquid chromatography 60 system (Thermo Fisher Scientific) was used for chromatographic separation and this was coupled to an Orbitrap HFX mass spectrometer (Thermo Fisher Scientific) via an EASY-Spray source. Electrospray nebulisation was achieved by interfacing to Bruker PepSep emitters (PN: PSFSELJ20, 20 μm, Bruker, Billerica, MA, USA). Peptide solutions were injected directly onto the analytical column (self-packed column, CSH C18 1.7 μm beads, 300 μm × 35 cm) at a working flow rate of 5 μL/min for 4 min. Peptides were separated using a 66-minute stepped gradient: 0–45% of buffer B (75% acetonitrile, 20% water, 5% DMSO with + 0.1% ferulic acid (FA)) for 66 min, followed by column re-conditioning and equilibration. Eluted peptides were analysed by the mass spectrometer in positive polarity, using an initial MS1 at 120,000 resolution followed by sequential MS2 acquisition and ion fragmentation at 30,000 resolution. An m/z range of 409.5 to 1650 was used. Raw data were processed using Spectronaut (v19.0[61]). Initial protein identification (pulsar search) allowed up to three missed cleavages and accounted for common protein modifications. Searches were conducted against the UniProt Homo sapiens 1-gene per protein sequence database (downloaded 22/01/2024, 20,596 entries) and a universal protein contaminants database[62] (downloaded 22/01/2024, 381 entries). For quantification, MS2-based data were analysed using a Direct DIA approach using the MaxLFQ method[63], and only proteotypic peptides were considered (q-value < 0.01). After removing entries from the protein contaminants database, further analyses were

performed using Perseus (v1.6.15.0[64]), where data were $\log_2$ transformed prior to additional filtering and statistical testing. For two-group comparisons, proteins with at least seven replicate intensities per experimental group were included, and a Student's t-test was applied with permutation-based FDR correction (q-value threshold of 0.05). Results were visualised as volcano plots, with significance defined by q-value. Perseus Gene Set Enrichment Analysis was performed, where intensities of t-test significant hits were z-score normalised and clustered using Hierarchical Clustering Analysis. Protein IDs within clusters were enriched using the Perseus module for Fisher's exact test, with multiple-testing correction by Benjamini-Hochberg (q-value threshold of 0.05), assessing the relative enrichment of GO terms per cluster against all background proteins. For Reactome enrichment analysis, UniProt ID mapping service was used to retrieve gene IDs prior to clusterProfiler (v4.12.0[65]) analysis, with a q-value threshold of 0.05.

**Triple quadrupole liquid chromatography-mass spectrometry (TQ LC-MS).** For intracellular drug concentration measurements, experiments were performed in 3 independent biological replicates. Cells were seeded into 96-well plates at 15000 cells/well and left to adhere overnight. Cells were pre-treated in quadruplicates for 1 h with or without bafilomycin in complete medium before being dosed with PARP inhibitors at their $IC_{50}$ concentrations. After 1 h of incubation, media were removed and cells were washed once in cold complete medium followed by one wash in cold Dulbecco's PBS. Cells were quenched in cold 80% (v/v) HPLC gradient grade acetonitrile in LC/MS grade water supplemented with 1.5 ng/mL Amisulpride-$d_5$ (CAY30075-1 mg) as an internal standard. Plates were sonicated for 3 min before supernatant was transferred to U-bottomed 96-well plates and centrifuged for 1 minute at 1000 x g. Extracted samples were stored at −80 °C. To calculate cell volume, parallel plates of untreated cells were counted using the Nexcelom Cellometer Auto Cell Counter (Nexcelom Bioscience, Lawrence, MA, USA) and cell diameter measurements were recorded.

Samples were thawed on ice and diluted 25 times in LC/MS grade water for analysis using the Waters Xevo® TQ-XS Mass Spectrometer coupled to an ACQUITY Premier System (Waters Ltd) using a CORTECS T3 Column, 120 Å, 2.7 μm, 2.1 mm×30 mm (186008481, Waters Ltd) at 60 °C. A 10 μL volume of sample was injected. Mobile phase A1 contained LC/MS grade water with 0.1% (v/v) formic acid and mobile phase B1 contained acetonitrile with 0.1% (v/v) formic acid. A gradient was applied over 3 min at a flow rate of 1.2 mL/min, beginning with a linear increase from 1% to 5% mobile phase B1 (0.00–0.50 min), followed by a gradient from 5% to 30% B1 (0.50–1.00 min), and a subsequent ramp from 30% to 100% B1 (1.00–1.50 min). The composition was then held at 100% B1 (1.50–2.40 min), returned to 1% B1 (2.40–2.50 min) and finally held at 1% B1 (2.50–3.00 min) to allow system re-equilibration. The sample eluate was injected into an Xevo® TQ-XS mass spectrometer (Waters Ltd) using electrospray ionisation in positive ion mode. MS conditions: source temperature 150 °C; capillary voltage 0.8 kV; desolvation temperature 600 °C; desolvation gas flow 1000 L/Hr; cone gas flow 150 L/Hr. System settings were controlled using MassLynx v4.2 software (Waters Ltd) and multiple reaction monitoring methods were optimised using IntelliStart and validated manually by direct infusion. Cone voltages (V) and collision energies (eV) were optimised for quantifier and qualifier mass transitions as follows: Amisulpride-$d_5$ quantifier 375.468 > 242.251 m/z (38 V, 28 eV), qualifier 375.468 > 196.237 m/z (38 V, 40 eV), retention time 0.86 min; rucaparib quantifier 323.984 > 293.05 m/z (34 V, 10 eV), qualifier 323.984 > 235.962 m/z (34 V, 32 eV), retention time 0.935 min; Niraparib quantifier 321.13 > 205.11 m/z (8 V, 42 eV), qualifier 321.13 > 304.14 m/z (8 V, 18 eV), retention time 1.0 min; Olaparib quantifier 435.18 > 281.3 m/z (10 V, 30 eV), qualifier 435.18 > 367.3 m/z (10 V, 20 eV), retention time 1.165 min. MassLynx v4.2 software

automatically calculated the dwell time for each transition with a minimum of 15 analytical points per peak. Samples were run alongside matrix-matched calibration curves generated using untreated cell extracts processed in parallel to the treated samples.

Data were processed using Skyline (v24.1) software[66]. Peak areas were normalised to the internal standard and plotted against the calibration curve before values were normalised to cell density measurements enabling intracellular concentration calculation. Values were normalised to PARP inhibitor only controls and data were plotted using GraphPad Prism (v10.4.1).

### Reporting summary

Further information on research design is available in the Nature Portfolio Reporting Summary linked to this article.

## Data availability

Mass spectrometry proteomics data have been deposited to the ProteomeXchange Consortium via the PRIDE partner repository with the dataset identifier PXD057265. Spatial transcriptomics data are available from GEO under accession code GSE281519. Remaining data are available within the Article, Supplementary Information, or Source Data file. Source data are provided with this paper.

## Code availability

Original code, including README files (used for modelling of spatial transcriptomic data) are available on Zenodo using the following link: https://doi.org/10.5281/zenodo.17610220.

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

## Acknowledgements

The authors would like to thank Hiromi Kudo and Robert Goldin for help with immunohistochemistry, Anke Nijhuis for providing cell lines, Nazma Malik for sharing the TFEB-GFP transient expression construct, Ka Lok Choi for assistance with genomics approaches, James Elliot and Joana de Teixeira Carrelha for assistance with flow cytometry, Oliver Gonzalez Carvajal and Mariana Veiga for assistance with cell collection, Aleksandra Gruevska for advice and assistance in mass spectrometry imaging, and Jesus Gil and Mariantonietta D'Ambrosia for their advice on the manuscript. We would also like to thank the patients who gave their consent for their material to be used in this study. Tumour tissues samples were provided by the Imperial College Healthcare NHS Trust Tissue Bank. Other investigators may have received samples from these same tissues. The research was supported by the National Institute for Health Research (NIHR) Biomedical Research Centre (BRC) based at Imperial College Healthcare NHS Trust and Imperial College London. The views expressed are those of the author(s) and not necessarily those of the NHS, the NIHR or the Department of Health. C.R.M. was supported by funding from the Integrative Toxicology Training Partnership administered via the MRC Toxicology Unit, awarded to L.F. and Z.H., and a Victoria's Secret Global Fund for Women's Cancers Career Development Award, in partnership with Pelotonia and AACR (24-20-73-FETS, L.F.). This work was also supported by funding from the Medical Research Council (MC-A564-5QC70, L.F. and MR/W019132/1, Z.H.) and a CRUK Career Establishment Award (RCCCEA-Nov21\100001, L.F.).

## Author contributions

Conceptualisation, C.R.M. and L.F.; Methodology, C.R.M., L.F., P.C., Z.H., and A.R.B.; Formal Analysis, C.R.M., G.Z., R.R., G.R.Y., A.M., D.M., and L.F.; Investigation, C.R.M., G.Z., P.O.P., E.D., K.T., L.P., P.V., and N.L.; Resources, C.W., I.A., B.P., B.R.P., V.W., Z.T., N.M., P.S., L.G., B.L., I.M., C.F., and A.R.B.; Writing (original draft), L.F.; Writing (review and editing), L.F., C.R.M., Z.H., P.C., C.F., B.L., and A.R.B.; Supervision, L.F. and Z.H.; Funding Acquisition, L.F. and Z.H.

## Competing interests

The Authors declare the following competing interests: I.M. has undertaken advisory boards for AstraZeneca, Clovis Oncology/pharma& and GSK. Imperial College has received institutional grant support from AstraZeneca. All other authors declare no competing interests.
