## [Transparent Peer Review file · Nature Communications]

Multimodal imaging reveals a lysosomal drug reservoir that drives heterogeneous distribution of PARP inhibitors

Corresponding Author: Dr Louise Fets

Version 0:

Reviewer comments:

Reviewer #1

(Remarks to the Author)

In this work, Moncayo et al. analyze the tissue, cellular, and subcellular distribution of several PARP inhibitors including niraparib, rucaparib, and olaparib. Two of these agents have pKa values in the physiological range, resulting in neutral and ionized states. Using a series of ex vivo clinical experiments and in vitro single cell experiments, the authors demonstrate single cell and tissue-level heterogeneity in drug uptake that correlates with response rates. Sorting experiments support that at the cellular level, many of these are epigenetic differences, and transcriptional analysis correlates with expression of many lysosomal associated genes. The level of drug uptake is correlated with drug response, showing a potential pathway for resistance. Overall, heterogeneity at the cellular and tissue level is understudied, and the use of mass-spec imaging and intrinsic fluorescence of rucaparib provides insight into the multiple factors driving this heterogeneity. Therefore, these results are of potential interest to the research community. There are major areas that would help support the hypotheses and improve the interpretability of the results. With this additional input, the results would be highly informative to the field.

The biggest concern overall is the lack of accounting for in vivo delivery. I appreciate the fact that the authors call this out by stating the vascular-independent effects, but in vivo delivery limitations could reverse the conclusions of the paper. I would hypothesize that this is not the case, but this is crucial for supporting the conclusions.

An experiment with xenograft or PDX tumors would demonstrate if there are delivery limitations for these agents. This could involve, for example, a xenograft tumor where the mouse is treated with rucaparib PO, IP, or IV delivery, and the tumor is resected. Adjacent slices could be imaged with or without additional ex vivo labeling to see if the high uptake regions correlate (e.g. MSI or fluorescence). Other experiments are possible, but this is one suggestion.

The fast diffusion of the small molecule likely results in fairly uniform delivery following slow oral absorption and daily dosing in patients. However, the ex vivo labeling used in all the current experiments raises the delivery issue that could be addressed with a small number of mice. The mouse experiment would also provide some insight into the absolute concentrations in a tumor xenograft at a given dose versus ex vivo staining.

To expand on the concern, I think delivery limitations are probably not likely (provided the time point isn't too close to the treatment time), but this is not guaranteed, and it could completely change the interpretation of the data. A lack of vascular delivery limitations is needed to support the hypothesis that the lysosomes act as a reservoir. If there are delivery limitations, for example, then the lysosomal trapping could hinder efficacy. In contrast, lysosomal trapping is beneficial if delivery is not a limitation by increasing retention/exposure as seen with ex vivo incubation. The single cell and ex vivo labeling experiments in the current work are not limited by delivery, so the translation to the in vivo scenario is not clear. For example, if only 1 million drug molecules reach a cell in vivo, a cell with 95% trapping would respond less than one with only 0% trapping, due to a 20-fold lower free drug concentration (not trapped in lysosomes) to interact with the target. If delivery is not an issue, then the free drug is the same for both cells (it equilibrates), but the one with 95% trapping has an additional 20-fold more intracellular drug for sustained response.

A second experiment that would significantly add to the results is testing the potency (IC50) of olaparib in the high and low rucaparib cells in Fig. 5g. The current results indicate that the olaparib inhibition may be less dependent on the rucaparib uptake (if the mechanisms of heterogeneity are different), supporting the use of alternative agents in heterogeneous/resistant populations.

Minor points

Please include the structure of all 3 drugs – this makes it more convenient for the reader to see the structural differences. The logP values don't vary dramatically, but this could be useful to include in addition to the pKa values.

Consider PDEx for Patient Derived Explants, since PDE models can sometimes mean Partial Differential Equation models to some pharmacokinetic researchers.

Page 2, line 74 – Please check the references throughout. The first reference looks like 15, although maybe it's 1-5?

Figure 5A – Why was GFP needed as a co-registering agent? Couldn't Hoechst be used with and without fixation? Was this cytosolic eGFP or one of the fusion proteins?

(Remarks on code availability)

Reviewer #2

(Remarks to the Author)

In their manuscript termed 'Multimodal imaging reveals a lysosomal drug reservoir that drives heterogeneous distribution of PARP inhibitors', Moncayo et al. study the heterogeneity in the uptake and response of different PARP inhibitors in clinical samples of high-grade serous ovarian carcinoma (HGSOC). Utilising patient derived explants, a multimodal imaging analysis and some in vitro validation of their data, the authors come to the conclusion that two out of the three drugs (Rucaparib and Niraparib, but not Olaparib), can be trapped in lysosomes, due to their nature as weak base, and that the lysosomes in turn act as a storage that releases the drugs, leading to increased efficacy.

This study is a methodological tour de force on several levels. It uses patient derived explants (PDE) for ex vivo screening of drug incorporation with downstream processing for mass spectrometric imaging (MSI), which offers a blueprint for similar approaches in other solid tumours. The utilisation of consecutive sections for the multimodal omics approaches is justified, given that consecutive sections give better quality data and that at a nominal resolution of 20 µm, the thickness of the sections is smaller than the lateral resolution of the MSI.

Overall, the findings are novel and justify the technologies used by the authors. The correlation of MSI and spatial transcriptomics truly allows to identify factors of heterogeneity between, and, crucially, within single tumours, and correlate this with drug uptake and efficacy. While some of the findings, such as lysosomal trapping itself, are not entirely new, in this context and with the added benefit of spatial multi-omics, they significantly advance our understanding of these tumours and potentially have a therapeutic impact. The authors contextualise their findings well and they strive to explain them with appropriate in vitro models, such as the sorting of RUC-high and -low cells to prove the transient nature of the lysosomal phenotype.

One observation that deserves some more discussion, is the distribution of the lysosomal heterogeneity in vivo. The authors describe the phenomenon of heterogeneity both in vitro and in vivo. Yet, in vivo it clearly represents a regional phenomenon, with many cells in certain tumour areas displaying the same behaviour. We do not quite reach single cell resolution, so we cannot determine, if within those areas some cells have more lysosomes than others, but in any case clear hot-spots are visible. As the authors are quite certain that these differences are not genetically encoded, it would be useful to at least speculate about the local driver for the divergent lysosomal content in different areas of the tumours.

Lastly, the data showing that both Rucaparib and Niraparib intracellular concentrations are decreased in tissue culture in the presence of bafilomycin while the concentration of Olaparib is not, is very interesting. However, it would be interesting to see, if the V-ATPase activator EN6 would instead increase levels of Rucaparib and Niraparib while, following the logic of these findings, Olaparib levels should stay unchanged.

Overall, this is a very sound study, and I deem it fit for publishing, after addressing the few points raised.

(Remarks on code availability)

I do not have the right expertise to review this code.

Reviewer #3

(Remarks to the Author)

This paper proposes an interesting concept that weak base PARP inhibitors (PARPi) (niraparib and rucaparib) accumulate in lysosomes providing a reservoir for PARPi that "determines" PARPi content in the nucleus. This could suggest that altering lysosome pH would alter the accumulation and action of some PARPi but not others. The manuscript is strengthened by the use of tissue slices from different sites in the peritoneum albeit from 3 patients as well as cell line models. The concordance between the slices and cell line models implicating lysosomes in accumulation of PARPi is considered a positive. There are several additional interesting points including marked interpatient, intertumoral (different tumors from same patient) and intratumoral concentrations of drug and associations with DNA damage and apoptosis. Further the differential accumulation of niraparib and potentially other PARPi is due to cell states and to underlying genetic aberrations, an important point. The authors conclude that altering lysosomes could alter efficacy to PARPi, a point that would only seem to apply to niraparib and rucaparib.

Specific comments

1. It would be useful to provide a table summarizing key pharmacologic properties of different PARPi (e.g. pKa, logD, molecular weight) and how they would be influenced by lysosomal accumulation. Talazoparib is an approved PARPi. Which

class does it belong to. Similar comments apply to next generation PARPi such as AZs PARP1 selective and brain penetrant models. These do not need to be demonstrated experimentally but should be presented and discussed.

2. The authors propose that lysosomes accumulate PARPi and that this determines nuclear levels. They demonstrate that nuclear levels correlate with lysosomal levels and that cells with high lysosomal PARPi have higher nuclear PARPi at least for niraparib. A key mechanistic step should be strengthened. Does the higher accumulation of PARPi in lysosomes result in higher PARPi activity. The associations are strong but do not seem definitive. Manipulation of lysosomal activity and changes in PARPi activity would strengthen the conclusions. Determining effects of chemical manipulation of lysosomes and in particular changing lysosomal content with TFEB manipulation and determining effects on PARPi activity and cell death such as colony formation would strengthen the conclusions.

3. To strengthen the claim that lysosomal content influences differential drug uptake, the authors should analyze publicly available bulk or single-cell RNA-sequencing datasets from patients treated with PARPi, particularly those receiving olaparib, rucaparib, or niraparib. Evaluating whether lysosome-associated genes are upregulated in responder patients would provide important supportive evidence for their hypothesis.

4. Given that the current study includes only 3 treatment-naïve patients, it would be particularly valuable to extend these drug exposure experiments to a larger and more diverse patient cohort that includes HR status and PARP responsive and resistant patients. In particular, including PARPi-resistant patients could offer important insights into whether lysosome-driven drug accumulation is preserved or altered in the context of endogenous or acquired resistance. However, this can be presented as a limitation of the current study rather than being required for the manuscript.

5. PEO4 and PEO1 cell lines originate from the same patient, with PEO4 known to exhibit greater resistance to PARPi, as also shown by the authors (Extended Data Fig 5). It is surprising that the authors did not perform a more detailed comparative analysis between these two cell lines. Such a comparison could provide valuable insights into whether the observed differences are primarily driven by lysosomal alterations or confounded by BRCA2 status. To minimize this, I recommend that the authors include additional analyses particularly linking lysosomal PARPi accumulation by directly comparing PEO1 and PEO4 cells. Additional cell lines with and without mutations in BRCA1/2 would also strengthen the case for accumulation of PARPi in lysosomes being of significance.

6. The authors need to add a section presenting the limitations of the study. These include those indicated above and some identified in the discussion but they should be described in more detail. These would include only 3 samples, a limited spectrum of PARPi, use of slices and cell lines rather than samples from treated patients, lack of correlation between the characteristics in the models and patient outcomes which relates directly to the paragraph on clinical relevance and others.

Minor issues

1. The manuscript refers to Figure 6g to demonstrate reduced intracellular drug levels in the presence of bafilomycin. However, Figure 6g appears to be missing from the current version of the manuscript. Based on the content, it seems likely that the authors intended to refer to Figure 6f instead. Please review the figures carefully, and if Figure 6g is indeed missing, ensure it is included in the revised submission.

(Remarks on code availability)

I am not able to review or run the code. This is not my expertise.

Reviewer #4

(Remarks to the Author)

(Remarks on code availability)

There were insufficient README files and instructions for running the code. Additionally, the files are restricted to users with special access. For reproducibility, the authors should make all code publicly available.

Version 1:

Reviewer comments:

Reviewer #1

(Remarks to the Author)

I appreciate the authors' efforts to collect in vivo data to support the conclusions of the paper. It is unfortunate that the technical limitations were unable to shed light on the in vivo distribution of these drugs. It is apparent that without tissue slicing, the millimeter-thick tumors did not have enough oxygen delivery to the center for ex vivo staining with 24 hrs in culture. It also appears, surprisingly, that the oral dosing was not able to achieve significant uptake despite the literature precedent for delivery at these doses. Potentially transporter differences (e.g. Murray et al. 2014, BJC), reduced interstitial

pH in the tumor, etc. could result in differential tumor uptake in this model.

Without the in vivo data to support a lack of tissue-level gradients, I feel this limitation should be expanded in the newly added section on limitations. It is unclear if the authors were using an orthotopic model (vs. subcutaneous), but the peripheral uptake from the IP delivery seems like it could be. The tissue-level gradients seen in this IP data actually seem to confirm the effect that I was mentioning. The high cellular uptake driven by lysosomal sequestration could be limiting the drug penetration from the surface of the tumor into the tumor core (given the apparent lack of sufficient tumor perfusion for vascular delivery in the center). Presumably, a reduction in lysosomal uptake (e.g. by BafA1, although it is not practical to do this experimentally) would reduce the signal in the tumor periphery and increase it in the tumor center. These are the tissue level (distance from blood vessels or from tumor surface) that I am focused on. This is different than the intracellular gradients that are focus of this paper. Therefore, while I really like the cellular imaging experiment showing BafA1 reduces lysosomal uptake and nuclear signal (indicating trapping doesn't prevent exposure within the same cell), it doesn't address the potential for tissue-level gradients, hence the need for the statement in the limitation. The authors mention the widespread clinical benefit for all 3 PARP inhibitors, which is why they didn't consider in vivo delivery limitations initially. I agree that these are unlikely in this particular case, but the same could be said for the cellular heterogeneity, where the drugs are successful but cellular heterogeneity can lead to differential patient responses.

Now, despite these comments, I still don't think the tissue-level gradients are an issue for this particular drug class. While tumors aren't necessarily exposed to the C_{max} plasma concentration due to blood flow limitations for small molecules, the high C_{min}/trough concentrations for these agents would prevent significant washout and long-term gradients from being established. However, these gradients could exist for other drugs (or for these drugs with different delivery vehicles), and without direct evidence for this case, I think it should be expanded in the limitation section. I think it's fine to include the statement that these gradients are unlikely to exist in this case given the high C_{min} and C_{max} concentrations, but these gradients could exist for other drugs (or delivery vehicles with these drugs) where lysosomal sequestration in the tumor periphery, for surface uptake, or perivascular cells, for vascular delivery, could limit tissue penetration. Known examples include different formulations of doxorubicin (which can lead to perivascular gradients and/or poor tumor uptake from liposomal formulations) and hypoxia-activated drugs. This would help put the current work in the broader context of heterogeneous drug delivery.

The other changes have addressed my other concerns and improved the clarity of the manuscript.

(Remarks on code availability)

Reviewer #2

(Remarks to the Author)

I have now assessed the revised manuscript 'Multimodal imaging reveals a lysosomal drug reservoir that drives heterogeneous distribution of PARP inhibitors' by Moncayo et al. The authors have put significant work into this revision and the manuscript is now in much better shape. My requests have been addressed. Some have not shown the expected results, but I appreciate the methodological limitations, which they discuss. Equally I think that addition of clarification in the discussion (lines 512-523) has greatly helped with the transparency of the work. Furthermore, the points raised by my fellow reviewers have been in many cases addressed, thus making the manuscript more complete. Overall I am now happy to recommend this work for publication. I regard it as an important addition in the field of Mass Spectrometric imaging based pharmacokinetic analysis.

(Remarks on code availability)

Reviewer #3

(Remarks to the Author)

We thank the authors for the extensive experimental data and experiments performed in response to concerns. It is disappointing that the in vivo studies were not successful in testing the hypotheses presented. The authors provide a rational for this concern which is reasonable. They mitigate the concerns somewhat by additional ex vivo and in vitro studies.

There are several in vitro/ex vivo additions that add to the paper. The short term treatment in the presence and absence of BAFA, the addition of PEO1 and PEO4 and additional cell lines all add to the manuscript and mitigate the majority of the concerns.

The manuscript is thus significantly strengthened.

(Remarks on code availability)

Reviewer #4

(Remarks to the Author)

(Remarks on code availability)

REVIEWER COMMENTS

Reviewer #1 (Remarks to the Author):

In this work, Moncayo et al. analyze the tissue, cellular, and subcellular distribution of several PARP inhibitors including niraparib, rucaparib, and olaparib. Two of these agents have pKa values in the physiological range, resulting in neutral and ionized states. Using a series of ex vivo clinical experiments and in vitro single cell experiments, the authors demonstrate single cell and tissue-level heterogeneity in drug uptake that correlates with response rates. Sorting experiments support that at the cellular level, many of these are epigenetic differences, and transcriptional analysis correlates with expression of many lysosomal associated genes. The level of drug uptake is correlated with drug response, showing a potential pathway for resistance. Overall, heterogeneity at the cellular and tissue level is understudied, and the use of mass-spec imaging and intrinsic fluorescence of rucaparib provides insight into the multiple factors driving this heterogeneity. Therefore, these results are of potential interest to the research community. There are major areas that would help support the hypotheses and improve the interpretability of the results. With this additional input, the results would be highly informative to the field.

The biggest concern overall is the lack of accounting for in vivo delivery. I appreciate the fact that the authors call this out by stating the vascular-independent effects, but in vivo delivery limitations could reverse the conclusions of the paper. I would hypothesize that this is not the case, but this is crucial for supporting the conclusions. An experiment with xenograft or PDX tumors would demonstrate if there are delivery limitations for these agents. This could involve, for example, a xenograft tumor where the mouse is treated with rucaparib PO, IP, or IV delivery, and the tumor is resected. Adjacent slices could be imaged with or without additional ex vivo labeling to see if the high uptake regions correlate (e.g. MSI or fluorescence). Other experiments are possible, but this is one suggestion. The fast diffusion of the small molecule likely results in fairly uniform delivery following slow oral absorption and daily dosing in patients. However, the ex vivo labeling used in all the current experiments raises the delivery issue that could be addressed with a small number of mice. The mouse experiment would also provide some insight into the absolute concentrations in a tumor xenograft at a given dose versus ex vivo staining.

To expand on the concern, I think delivery limitations are probably not likely (provided the time point isn't too close to the treatment time), but this is not guaranteed, and it could completely change the interpretation of the data. A lack of vascular delivery limitations is needed to support the hypothesis that the lysosomes act as a reservoir. If there are delivery limitations, for example, then the lysosomal trapping could hinder efficacy. In contrast, lysosomal trapping is beneficial if delivery is not a limitation by increasing retention/exposure as seen with ex vivo incubation. The single cell and ex vivo labelling experiments in the current work are not limited by delivery, so the translation to the in vivo scenario is not clear. For example, if only 1 million drug molecules reach a cell in vivo, a cell with 95% trapping would respond less than one with only 0% trapping, due to a 20-fold lower free drug concentration (not trapped in lysosomes) to interact with the target. If delivery is not an issue, then the free drug is the same for both cells (it equilibrates), but the one with 95% trapping has an additional 20-fold more intracellular drug for sustained response.

We thank the reviewer for their fair and constructive assessment of our work, helpful suggestions, and for highlighting that the area is understudied and that our findings could be 'highly informative to the field'.

We appreciate that our initial submission did not include studies with *in vivo* delivery: as the reviewer points out, we were deliberately focused on the cell intrinsic properties of drug accumulation as this is a particularly under-studied area, and for these purposes, we see our vascularity-independent approach as a benefit. Furthermore, given the widespread clinical benefit established for all three PARP inhibitor drugs we investigated, we did not initially consider possible *in vivo* delivery limitations. Finally, we picked the concentrations used for our ex-vivo incubations to be in a similar range to the C_{max} observed in patient studies (from methods section 'Explants were treated with niraparib, olaparib and rucaparib at 20 μ M in complete medium or with 0.1% DMSO as vehicle. PARP inhibitors have a C_{max} (source: drugbank) in the micromolar range (12.4-17.5 μ M, 6 μ M and 2.5 μ M for olaparib, rucaparib and niraparib respectively), therefore the concentrations used for explants are in line with those observed in patients').

We do however agree with the reviewer that addition of *in vivo* studies would add weight to our work, and so we set out to address this with mouse work, combined with additional *in vitro* studies (outlined below) to address the reviewers' query on reversal of the 'reservoir' to a 'trap' at more limiting concentrations of rucaparib. Although the *in vivo* work was not successful and revealed that substantial optimisation would be necessary, we believe our *in vitro* work has successfully addressed the reviewers concern.

In vivo work:

We chose to use the well-established ID8 mouse model, with the rationale that this orthotopic, syngeneic tumour type would best model the tumour microenvironment and heterogeneity of human disease. Unfortunately, the reviewer's suggestion of using an adjacent slice for comparison with *ex vivo* dosing would not be technically feasible since our patient-derived explant slices are 400-600 μ m thick, meaning tissue architecture is altered substantially between slices, and importantly, this would involve re-dosing a pre-*in vivo*-dosed tumour, which could confound interpretation.

Therefore, we addressed the question in the following way: ID8 syngeneic ovarian tumour-bearing mice were dosed daily for 5 days with rucaparib, either *per os* (100 mg/kg) or *intra peritoneally* (10 mg/kg) to enable steady state drug accumulation, and tumours were harvested 2 hours after rucaparib dosing on the 5th day. Doses of rucaparib were chosen based on concentrations previously found to be effective in mice dependent on route of administration¹. We then compared drug distribution in tumours from these dosed mice to tumours (obtained in parallel) from un-dosed mice that were subsequently incubated with rucaparib *ex vivo* (20 μ M for 24h) to recapitulate our PDE protocol. We were conscious that tumour vascularity, interstitial fluid pressure and cell composition (e.g., stromal and immune infiltration) will all influence drug accumulation and may differ significantly between animals. We therefore compared drug distribution across 5 *in vivo* dosed tumours to 5 *ex vivo* dosed, to take account of this variability and draw conclusions on comparability of the methods.

The data demonstrated quite different drug distributions between the *ex vivo* dosed, *IP* dosed and *per os* dosed routes (despite using known effective concentrations of rucaparib for the two latter routes¹, **Reviewer Figure 1a**). The *IP* tumours clearly accumulated rucaparib, however this was concentrated around the tumour periphery, whereas rucaparib *per os* was below the detection limit. These results, despite 5 days of dosing, in an attempt to reach steady-state drug levels, may have resulted from numerous reasons including sub-optimal dosing, drug metabolism, or poor vascularisation within the tumour model. Indeed, limited vascularity was reflected by limited CD31 staining in the majority of tumours.

In contrast, we saw substantial rucaparib signal in *ex-vivo* dosed tumours, however we are cautious in our interpretation of this result, and particularly in comparison between this and the *in vivo*-dosed models. This is because strong eosin staining in H&E sections demonstrated that the cores of these tumours showed evidence of necrosis, which was not evident in the *in vivo* dosed tumours. The small size of these tumours (less than 1 mm in thickness) prevented slicing after removal from the animal; therefore, they were cultured whole. The necrotic core suggests that the protocol used to our PDE samples did not adapt well to the mouse tumours. This necrosis was also likely the reason for widespread staining in the CD31 channel only in the *ex vivo* dosed tumours, which arose from non-specific binding of the secondary antibody, only in these samples (**Reviewer Figure 1b**). Although the rucaparib signal in these *ex vivo* dosed samples is specific (**Reviewer Figure 1c**), drug perfusion through necrotic tissue would be very different to healthy tumour, therefore it is not appropriate to compare drug distribution in the two models since tissue architecture is not comparable. It is important to note that in our patient-derived explant model, which has been extensively optimised by our collaborators to ensure viability in culture for up to 5 days, we did not see any evidence for necrosis.

Given the lack of comparability between conditions, we feel that this data would not add weight to the manuscript. While we agree that *in vivo* validation would be of benefit, this experiment has demonstrated that answering this question will require substantial optimisation, including use of a new *in vivo* model, and optimisation of *ex vivo* culture conditions for mouse tumours, before meaningful comparison could be made to our patient-derived samples. Given our stated focus on the cell-intrinsic heterogeneity of drug distribution, we feel that the level of optimisation that would be required goes beyond the scope of this manuscript.

In vitro work:

The reviewer's suggestion of the role of the lysosome changing from a reservoir to a trap dependent upon extracellular drug availability is an excellent point, and one we felt compelled to address, particularly given that our *in vivo* work was unable to provide a conclusive response. Based on the LogD model, our assumption would be that the level of drug that accumulates within the lysosome would be proportional to the amount of drug entering the cell, and the percentage protonation would be dependent on lysosomal pH and independent of intracellular/extracellular concentration. This would mean that when more drug is available extracellularly, more drug accumulates in the lysosome, and at lower levels (such as in poorly vascularised regions), less will accumulate, however the relative cytosolic: lysosomal ratio would remain constant.

To clarify this important point, we used an imaging-based approach to measure the nuclear level of rucaparib (as a read out of the relative level available to bind PARP1/2) at concentrations both above and below our previously tested IC₅₀ concentration (8.5 µM). Using

1 μM extracellular rucaparib, to mimic regions of tumour with reduced delivery, prevention of lysosomal accumulation using bafilomycin A1 (BafA1) still decreased the levels of rucaparib in the nucleus, suggesting the lysosome was acting as a reservoir, rather than a 'trap' even at these low concentrations (**Reviewer Figure 2/ Supplementary Data Figure 9I**). This effect was maintained up to 15 μM . Importantly, the relative decrease was consistent between vehicle and BafA1-treated cells at the different concentrations, supporting the model that it is the percentage protonation of rucaparib that drives the BafA1-mediated effect, and this is independent of absolute concentration (**Reviewer Figure 2/ Supplementary Data Figure 9I**). This data supports our conclusions, with respect to the applicability of our findings to lysosomal accumulation for *in vivo* dosed tumours, where local drug delivery may be variable or limiting. It has been added to the manuscript Supplementary Data Figure 9I and is discussed in **lines 410-416**.

We believe the *in vitro* experiment above addresses the reviewer's concern; however we have added in the discussion a paragraph describing the limitations of our work, which includes the lack of studies *in vivo*. We also reiterate that differences in vascularity will mean that heterogeneity of PARP inhibitor distribution will likely be even more substantial than we have observed in our study (**lines 550-554**).

a

Scale bar = 1mm

Reviewer Figure 1. a) Comparative distribution of rucaparib in ID8 ovarian tumours following *in vivo* (IP/OS) and *ex vivo* dosing regimens. ID8 syngeneic ovarian tumour-bearing mice were administered rucaparib daily for five days via two routes: intra-peritoneal (IP; 10 mg/kg) or per os (OS; 100 mg/kg) (n=5 per group). Animals were culled 2 hours post-last dose, and tumours were collected. For the *ex vivo* group (n=7), tumours from untreated animals were extracted, cultured overnight on cell culture inserts, and then incubated with rucaparib (20 µM, n = 5) or vehicle-only (negative control, n = 2) for 24 hours. All tumours were swiftly washed, snap-frozen, and stored before being embedded in HPCM+PVP and subsequently cryosectioned (12 µm). Prior to imaging or staining, tissues were thawed under vacuum to prevent water condensation and analyte delocalization. Adjacent sections were processed for the following three imaging modalities: H&E; CD31/WGA (green: anti-CD31 antibody, ab28364, 1:200 dilution; orange: WGA-594 conjugate, Biotium's 29023-1, 1:1000 dilution); and rucaparib. Rucaparib was imaged in un-fixed tissue and with no counterstain to avoid fluorescence quenching and/or analyte delocalisation. All images were acquired using a ZEISS Axioscan Z1 slide scanner at 20x magnification. **b)** Control samples demonstrating non-specific secondary antibody staining in *ex vivo* dosed tumours, but not *in vivo* dosed tumours. **c)** Control samples demonstrating the specificity of rucaparib fluorescence in *ex vivo* dosed samples.

Reviewer Figure 2. PEO1 cells were pre-treated with BafA1 for 1 hour, followed by treatment with vehicle or Rucaparib (1, 5, or 15 μM) for an additional hour, in the presence or absence of BafA1. Imaging-based quantification of nuclear levels of rucaparib per cell was carried out as before (left, representative of $n = 3$ biological replicates). Median rucaparib per nucleus was calculated per biological replicate and normalised to matched vehicle control (right). 1-way and 2-way ANOVA and Sidak multiple comparison test were applied to evaluate statistical significance respectively for left and right panels.

A second experiment that would significantly add to the results is testing the potency (IC_{50}) of olaparib in the high and low rucaparib cells in Fig. 5g. The current results indicate that the olaparib inhibition may be less dependent on the rucaparib uptake (if the mechanisms of heterogeneity are different), supporting the use of alternative agents in heterogeneous/resistant populations.

We agree that based on our model, rucaparib-high cells should not have differential sensitivity to olaparib, compared to rucaparib-low cells. However, this hypothesis is difficult to test in an unbiased way, since if we were to sort ruc-high and low cells, ruc-high cells would have decreased viability as a result of the initial exposure (as shown in Figure 5f, g). Therefore, we addressed this question using cells with high lysotracker staining as a proxy for high-rucaparib, since the two are very highly correlated (Figure 4 e, f). After sorting lysotracker-high and lysotracker low cells (**Reviewer Figure 3 a/ Supplementary Data Figure 10 b**), they were placed back into culture overnight to re-adhere prior to treatment. Unfortunately, these cell populations rapidly re-equilibrate their lysosomal content, as shown when re-incubating with lysotracker the following day, and this change mirrors the levels of rucaparib that accumulate in these different cell populations (**Reviewer Figure 3 b/ Supplementary Data Figure 10 c**). Despite the diminished differences in lysosomal content between populations, we found that upon challenge of lysotracker-high and low cell populations with either rucaparib or olaparib, we saw differential responses, as predicted by our model. While the levels of γH2AX staining were significantly increased in lysotracker-high treated cells challenged with rucaparib compared to lysotracker-low cells, there was no difference between the γH2AX signal in the two populations when challenged with olaparib (**Reviewer Figure 3 c/ Figure 6 g**). This supports the hypothesis that heterogeneity in accumulation of the different drugs results from different mechanisms. This data has been added in as Figure 6 g and Supplementary Data Figure 10 b and c, and is discussed in **lines 441-451**.

Reviewer Figure 3. a) $Lyso_{High}$ and $Lyso_{Low}$ PEO1 cells (top and bottom 20% of lysotracker-treated population respectively) were FACS-sorted with total lysotracker and vehicle-only controls after a 30 min treatment. Sorted cells were re-cultured overnight in drug-free media. **b)** Intracellular rucaparib and lysotracker levels were assessed by incubating cells with drug or lysotracker for 1 hour prior to FACS analysis ($n = 2$ biological replicates for lysotracker). For a and b, unless otherwise stated, data depicts $n=3$ biological replicates with $n=3$ technical replicates. Two - way ANOVA and Sidak multiple comparison test were applied to evaluate statistical significance. **c)** $Lyso_{High}$ and $Lyso_{Low}$ PEO1 cells (top and bottom 20% of lysotracker-treated population respectively) were FACS-sorted, re-cultured overnight in drug-free media and treated at IC_{50} levels of Rucaparib, Olaparib, or vehicle for 24 hours. DNA damage was assessed by quantifying number of γ H2AX foci relative to matched vehicle controls per replicate. Data depicts $n=3$ biological replicates with $n=3$ technical replicates. Two-way ANOVA and Sidak multiple comparison test were applied to evaluate statistical significance.

Minor points

Please include the structure of all 3 drugs – this makes it more convenient for the reader to see the structural differences. The logP values don't vary dramatically, but this could be useful to include in addition to the pKA values.

We have added the structures of all three PARP inhibitors to **Figure 1a**, and an additional table (**Supplementary Data Table 2**) which shows both the structure and a number of relevant chemical features, including LogP. To address one of Reviewer 3's points, we have included the properties of several other PARP inhibitors which have been used in clinical trials, which allows us to compare properties and hypothesise lysosomotropism at the end of the manuscript.

Consider PDEx for Patient Derived Explants, since PDE models can sometimes mean Partial Differential Equation models to some pharmacokinetic researchers.

While we understand the reviewer's concern, we feel that PDE is a widely used acronym for Patient Derived Explants, and given the broad readership of Nature Communications, changing to PDEx could lead to confusion. We have ensured that this acronym is well defined both in the text and in figure legends.

Page 2, line 74 – Please check the references throughout. The first reference looks like 15, although maybe it's 1-5?

We thank the reviewer for alerting us to this – it has now been corrected.

Figure 5A – Why was GFP needed as a co-registering agent? Couldn't Hoechst be used with and without fixation? Was this cytosolic eGFP or one of the fusion proteins?

We used cytosolic GFP as a co-registration marker because its excitation and emission spectra do not overlap with rucaparib, unlike Hoechst. The GFP is not fused to any protein.

Reviewer #2 (Remarks to the Author):

In their manuscript termed 'Multimodal imaging reveals a lysosomal drug reservoir that drives heterogeneous distribution of PARP inhibitors', Moncayo et al. study the heterogeneity in the uptake and response of different PARP inhibitors in clinical samples of high-grade serous ovarian carcinoma (HGSOC). Utilising patient derived explants, a multimodal imaging analysis and some in vitro validation of their data, the authors come to the conclusion that two out of the three drugs (Rucaparib and Niraparib, but not Olaparib), can be trapped in lysosomes, due to their nature as weak base, and that the lysosomes in turn act as a storage that releases the drugs, leading to increased efficacy.

This study is a methodological tour de force on several levels. It uses patient derived explants (PDE) for ex vivo screening of drug incorporation with downstream processing for mass spectrometric imaging (MSI), which offers a blueprint for similar approaches in other solid tumours. The utilisation of consecutive sections for the multimodal omics approaches is justified, given that consecutive sections give better quality data and that at a nominal

resolution of 20 μm , the thickness of the sections is smaller than the lateral resolution of the MSI.

Overall, the findings are novel and justify the technologies used by the authors. The correlation of MSI and spatial transcriptomics truly allows to identify factors of heterogeneity between, and, crucially, within single tumours, and correlate this with drug uptake and efficacy. While some of the findings, such as lysosomal trapping itself, are not entirely new, in this context and with the added benefit of spatial multi-omics, they significantly advance our understanding of these tumours and potentially have a therapeutic impact. The authors contextualise their findings well and they strive to explain them with appropriate *in vitro* models, such as the sorting of RUC-high and -low cells to prove the transient nature of the lysosomal phenotype.

One observation that deserves some more discussion, is the distribution of the lysosomal heterogeneity *in vivo*. The authors describe the phenomenon of heterogeneity both *in vitro* and *in vivo*. Yet, *in vivo* it clearly represents a regional phenomenon, with many cells in certain tumour areas displaying the same behaviour. We do not quite reach single cell resolution, so we cannot determine, if within those areas some cells have more lysosomes than others, but in any case clear hot-spots are visible. As the authors are quite certain that these differences are not genetically encoded, it would be useful to at least speculate about the local driver for the divergent lysosomal content in different areas of the tumours.

We are very grateful to the reviewer for their insightful suggestions to improve the manuscript, and for highlighting that our study is a 'methodological tour de force', the results of which 'significantly advance our understanding of these tumours'. The reviewer is correct that lysosomal accumulation of drugs has been observed previously, however to our knowledge, ours is the first study that demonstrates this for the PARP inhibitors rucaparib and niraparib.

The reviewer is correct to point out the clear hotspots of drug in our *ex vivo* samples, and we agree that it is very interesting to speculate what could be driving this divergent lysosomal content. We should clarify (and have now added to the discussion, **lines 519-523**) that although we have evidence that the heterogeneity observed *in vitro* is not genetically encoded, we do not have the appropriate data to conclude this in the PDE samples. Lysosomal biogenesis is regulated by a number of transcription factors, including TFEB, the activity of which can be linked to several metabolic signalling pathways, such as a decrease in mTOR signalling or an increase in AMPK signalling²⁻⁴. Increase in cGAS/STING signalling, whose activation can be driven by copy number instability and micronuclei formation⁵, which is known to demonstrate intra-tumoural heterogeneity in ovarian cancer⁶, can also lead to increases in lysosome biogenesis⁷.

We explored transcriptional signatures for these factors within our high and low drug regions, to determine whether any of these factors could contribute to the observed lysosomal signature differences (**Reviewer Figure 4**). We could not see any significant differences within the pathways, other than increased mTOR pathway signatures in both high-rucaparib and high niraparib ROIs. Since decreased mTOR activity leads to lysosomal biogenesis, this is the opposite of what would be expected, but does fit with the fact that these regions were enriched with signatures associated with high metabolic activity, particularly glycolysis (Supplementary Data Fig 4c). In this instance, we feel that the lack of statistically significant associations does not mean that these potential drivers of lysosomal biogenesis should be ruled out, as it is likely

that our limited sample size and ROI-, rather than single-cell, based spatial transcriptomics approach do not afford us the power to properly test this interaction. As such, we have decided not to include the data in Reviewer Figure 4, but have added a paragraph to the discussion (lines 512-523) that speculates on the drivers of the lysosomal heterogeneity we observe.

Reviewer Figure 4. Boxplots to illustrate gene set scores between high- and low-drug regions in rucaparib and niraparib-treated PDE samples, extracted from GeoMx spatial transcriptomics data. Gene sets relevant to AMPK and mTOR signalling were obtained primarily from the Molecular Signatures Database (MSigDB) via the GSEA resource. Specifically, the Reactome mTOR–AMPK and AMPK regulation activity gene sets, the GOBP CAMKK_AMPK_SIGNALING_CASCADE (AMPK signalling) gene set, and the BioCarta mTOR pathway set were included. In addition, the CIN70 signature was obtained from Carter et al. (Nature Genetics, 2006). Gene set variation analysis (GSVA) was performed to compute enrichment scores for each spatial region of interest (ROI) within the spatial transcriptomics dataset, providing a quantitative estimate of pathway activity with spatial resolution. The statistical comparison incorporates patient and slice structure and was performed using an ANOVA for likelihood ratio test between a null mixed-effects model (gene-score $\sim (1|Patient) + (1|Patient:Slice)$) and a model including drug-response status (gene-score $\sim \text{high-low-status} + (1|Patient) + (1|Patient:Slice)$). Thus, the reported *p*-values are corrected based on patient and slices variability rather than simple direct testing between the high or low drug status.

Lastly, the data showing that both Rucaparib and Niraparib intracellular concentrations are decreased in tissue culture in the presence of bafilomycin while the concentration of Olaparib is not, is very interesting. However, it would be interesting to see, if the V-ATPase activator

EN6 would instead increase levels of Rucaparib and Niraparib while, following the logic of these findings, Olaparib levels should stay unchanged.

We thank the reviewer for suggesting this interesting experiment, and have completed LC-MS drug uptake analyses to this end using both PEO1 and OVCAR4 cell lines. While we could see a clear increase in niraparib but not olaparib in PEO1 cells treated with EN6, relative to control samples (fitting the predicted response; **Reviewer Figure 5**), there was only a subtle, albeit significant change in rucaparib when co-dosed with EN6. In OVCAR4 cells, we see no significant shift in the intracellular concentration of any of the drugs when co-treated with EN6.

We believe that there may be two reasons for these results. The first reflects the LogD minima of both niraparib and rucaparib, which plateaus at approximately pH 5 (Figure 6e). The pH of lysosomes is approximately 4.5-5, a value that may vary slightly from one cell line to another, dependent on expression levels of factors such as the v-ATPase. Therefore, if the lysosomal pH in OVCAR4 is already close to, or lower than the point at which the rucaparib or niraparib LogD has reached a minima, this could mean that % protonation of the drugs are minimally affected by further activation of the v-ATPase by EN6, and thus there is little or no change in drug accumulation. The second reason is technical; the relative noise level associated with our LC-MS measurements is substantially higher than the FACS-based approaches used to investigate rucaparib. Since the relative increase in rucaparib concentration elicited by EN6 treatment as observed by FACS (Figure 4g) is substantially smaller than the decrease caused by BafA1 (supporting reason one above), we may simply not have the power to detect a significant change with our LC-MS based approach. Since we cannot rule out a technical issue here, we believe that it would be better not add this data to the manuscript.

Reviewer Figure 5. Intracellular concentration of olaparib, niraparib and rucaparib in PEO1 and OVCAR4 cells with or without treatment with EN6 (50 μ M) to acidify lysosomes. Data represents combined data from 4 (PEO1) or 3 (OVCAR4) biological replicates, each within 4 technical replicates, and are plotted relative to PARP inhibitor only control, within biological replicate. Statistical significance assessed by 2-way ANOVA with Dunnet's correction for multiple comparisons.

Reviewer #3 (Remarks to the Author):

This paper proposes an interesting concept that weak base PARP inhibitors (PARPi) (niraparib and rucaparib) accumulate in lysosomes providing a reservoir for PARPi that “determines” PARPi content in the nucleus. This could suggest that altering lysozyme pH would alter the accumulation and action of some PARPi but not others. The manuscript is strengthened by the use of tissue slices from different sites in the peritoneum albeit from 3 patients as well as cell line models. The concordance between the slices and cell line models implicating lysosomes in accumulation of PARPi is considered a positive. There are several additional interesting points including marked interpatient, intertumoral (different tumors from same patient) and intratumoral concentrations of drug and associations with DNA damage and apoptosis. Further the differential accumulation of niraparib and potentially other PARPi is due to cell states and to underlying genetic aberrations, an important point. The authors conclude that altering lysosomes could alter efficacy to PARPi, a point that would only seem to apply to niraparib and rucaparib.

1. It would be useful to provide a table summarizing key pharmacologic properties of different PARPi (e.g. pKa, logD, molecular weight) and how they would be influenced by lysosomal accumulation. Talazoparib is an approved PARPi. Which class does it belong to. Similar comments apply to next generation PARPi such as AZs PARP1 selective and brain penetrant models. These do not need to be demonstrated experimentally but should be presented and discussed.

We thank the reviewer for their positive summary of our work, and their very helpful suggestions to improve our manuscript. We have added a supplementary table (**Supplementary Table 2**) that describes the properties of all of the PARP inhibitors tested, plus talazoparib, veliparib, saruparib (AZD5305), pamiparib and palacaparib (AZD9574), and have added a paragraph referencing the properties of these drugs and their likely interaction with lysosomes in the discussion (**lines 525-530**).

2. The authors propose that lysosomes accumulate PARPi and that this determines nuclear levels. They demonstrate that nuclear levels correlate with lysosomal levels and that cells with high lysosomal PARPi have higher nuclear PARPi at least for niraparib. A key mechanistic step should be strengthened. Does the higher accumulation of PARPi in lysosomes result in higher PARPi activity. The associations are strong but do not seem definitive. Manipulation of lysosomal activity and changes in PARPi activity would strengthen the conclusions. Determining effects of chemical manipulation of lysosomes and in particular changing lysosomal content with TFEB manipulation and determining effects on PARPi activity and cell death such as colony formation would strengthen the conclusions.

We thank the reviewer for this point, and fully agree with them that perturbation experiments directly linking lysosomal content and/or pH to PARP inhibitor activity would strengthen our mechanistic conclusions. These approaches are complicated by the fact that manipulation of lysosomal biogenesis or function for extended periods, also has an impact on autophagy, as the two things are tightly linked by the CLEAR network of genes that are targeted by lysosomal

biogenesis transcription factors such as TFEB. Since autophagy itself influences DNA damage response⁸⁻¹⁰, it is therefore difficult to deconvolve changes in PARP inhibitor response from changes in autophagy.

To overcome this issue, we designed a short-term, 1-hour ‘drug-loading’ experiment. Cells were exposed to PARP inhibitors in the presence or absence of BafA1, after which the drug and BafA1 were removed to minimise any effects on autophagy (which is activated following the DNA damage response, and would be inhibited by extended BafA1 treatment), and cells were allowed to proliferate for 72 hours. The rationale is that for lysosomotropic drugs (Niraparib and Rucaparib only), the lysosome acts as a temporary reservoir: once the drug is removed from the media, the intracellular lysosomal store can sustain PARP1 engagement and trigger cytotoxicity. In this setup, BafA1 should have a protective effect by preventing lysosomal accumulation during the 1-hour pulse without the autophagy confound. Indeed, we saw that at increasing concentrations of rucaparib and niraparib, viability of cells was significantly reduced relative to those that had been co-treated with BafA1 to prevent lysosomal accumulation (**Reviewer Figure 6/ Figure 6 h, i**). This was however not the case for olaparib, further supporting our model that it does not accumulate in the lysosome. This data has been added to Figure 6h and i, and the experiment is described in the text in **lines 453-463**.

Reviewer Figure 6. a) Schematic illustrating the effect of V-ATPase inhibition on intracellular concentrations of weakly basic drugs. Under normal conditions, lysosomal acidification drives drug accumulation within lysosomes, creating a reservoir that sustains intracellular drug levels and target engagement after extracellular drug is removed. Inhibition of V-ATPase with BafA1 prevents lysosomal acidification, reducing both lysosomal and total intracellular drug levels. **b)** PEO1 cells were exposed to PARP inhibitors for 1 hour in the presence or absence of BafA1. Following treatment, both drug and BafA1 were removed, and cells were re-cultured in drug-free media for 72 hours to assess downstream effects. Data depicts n=3 biological replicates with n=3 technical replicates. Statistical significance was assessed by 2-way ANOVA with Dunnet’s correction for multiple comparisons.

3. To strengthen the claim that lysosomal content influences differential drug uptake, the authors should analyze publicly available bulk or single-cell RNA-sequencing datasets from patients treated with PARPi, particularly those receiving olaparib, rucaparib, or niraparib. Evaluating whether lysosome-associated genes are upregulated in responder patients would provide important supportive evidence for their hypothesis.

We fully agree with the reviewer that this would be a fascinating analysis that would strengthen our findings, but to the best of our knowledge RNA sequencing data from patients treated with rucaparib is not available, either in bulk or single cell. We have gone through the literature, and have consulted with both Professor Iain McNeish (a collaborator on this manuscript) and Professor Elizabeth Swisher, both of whom have led ARIEL and other clinical studies on rucaparib usage in High Grade Serous Ovarian Carcinoma. Data from these studies has focused on genomic rather than transcriptomic response markers, which is therefore unfortunately not suitable. Rucaparib has been in clinical use a relatively short time (FDA approval from December 2016, EMA approval from May 2018), therefore we hope that as usage expands that this data may become available to allow us to complete this sort of analysis at a later date. We have added a line acknowledging that this type of analysis, when data becomes available, would further strengthen the clinical relevance of this work (**lines 544-546**)

4. Given that the current study includes only 3 treatment-naïve patients, it would be particularly valuable to extend these drug exposure experiments to a larger and more diverse patient cohort that includes HR status and PARP responsive and resistant patients. In particular, including PARPi-resistant patients could offer important insights into whether lysosome-driven drug accumulation is preserved or altered in the context of endogenous or acquired resistance. However, this can be presented as a limitation of the current study rather than being required for the manuscript.

We also agree with the reviewer that it would be very beneficial to look into more diverse patient populations, and we plan to implement this in future. We have highlighted in **lines 538-543** that the current low number of patient samples is a limitation, and expanding to a more diverse cohort will be informative to offer additional clinical insights.

5. PEO4 and PEO1 cell lines originate from the same patient, with PEO4 known to exhibit greater resistance to PARPi, as also shown by the authors (Supplementary Data Fig 5). It is surprising that the authors did not perform a more detailed comparative analysis between these two cell lines. Such a comparison could provide valuable insights into whether the observed differences are primarily driven by lysosomal alterations or confounded by BRCA2 status. To minimize this, I recommend that the authors include additional analyses particularly linking lysosomal PARPi accumulation by directly comparing PEO1 and PEO4 cells. Additional cell lines with and without mutations in BRCA1/2 would also strengthen the case for accumulation of PARPi in lysosomes being of significance.

PEO1 cells were originally derived from a cisplatin-sensitive ovarian cancer patient and were later shown to carry a *BRCA2* mutation that produces a truncated, non-functional protein¹¹. After relapse, PEO4 cells were established from the same patient. These cells acquired a secondary, so-called 'reversion', *BRCA2* mutation that restores the open reading frame and protein function, thereby conferring cisplatin resistance. Because both PARP inhibitors and cisplatin act by inducing DNA damage, drug sensitivity in these models is primarily thought to be dictated by homologous recombination (HR) capacity: *BRCA2*-deficient cells (PEO1) are HR deficient, and therefore sensitive, whereas *BRCA2*-restored cells (PEO4) are HR-proficient and therefore resistant. We do however agree that it would be interesting to

understand whether differential lysosomal content or accumulation factors into this resistance, and as such, we have measured intracellular rucaparib levels in both PEO1 and PEO4, in the presence and absence of BafA1, to understand the extent of rucaparib accumulation that is driven by lysosomal sequestration. We found that contrary to the increased resistance observed in PEO4, they actually accumulated more rucaparib than PEO1 cells, and that this can be displaced by BafA1 treatment (**Reviewer Figure 7 a**). The increase in rucaparib was reflected by a concomitant increase in lysosomal content (**Reviewer Figure 7 b**). This data suggests that the resistance observed PEO1 is due to regained HR proficiency, and is not contributed to by changes in overall drug accumulation or lysosomal sequestration. To add to this, we tested sensitivity of PEO1 and PEO4 to niraparib and rucaparib, which are lysosomotropic, as well as olaparib and cisplatin, which are not. All drugs are known to be more effective in HR deficient cells, and in all cases, PEO4 were substantially more resistant (**Reviewer Figure 7 c**). Together, this data suggests that the major driver of increased rucaparib resistance in PEO4 is the regained HR proficiency, and alterations to lysosomal content and function do not have a major effect.

In addition, we examined rucaparib distribution across a range of cell lines from different cancer types, and assessed the proportion of signal accounted for lysosomal accumulation by treating all cells with either vehicle, or BafA1. This data demonstrates that lysosomal accumulation accounts for 60-80% of rucaparib signal, regardless of cancer type (**Reviewer Figure 7 d/ Figure 4 h**).

We have added the PEO1 and PEO4 comparison data to this figure, and it has now been added to the main text as Figure 4 h, discussed in **lines 358-367**.

Reviewer Figure 7. a and b) FACS quantification of rucaparib levels and lysotracker signal in PEO1 and PEO4 cell lines, in the presence and absence of BafA1 to prevent lysosomal accumulation. **c)** Dose response curves for PEO1 and PEO4 with different DNA-damage inducing drugs. IC₅₀ values were obtained in 3 independent biological replicates. (Cisplatin 0.3 ± 0.1 and 3.4 ± 0.3 μM; Niraparib 6.1 ± 1.8 and 23.5 ± 4.2 μM; Olaparib 2.6 ± 0.5 and 18.8 ± 2.2 μM; Rucaparib 9.0 ± 1.5 and 35.4 ± 1.7 μM for PEO1 and PEO4 respectively). *P*-value was ≤ 0.0001 for all comparisons in extra-sum-of-squares F test. **d)** Different cell lines were treated with rucaparib (10 μM) for 1 hour in the presence or absence of BafA1. Lysosomal rucaparib accumulation was estimated by calculating the fold-change in drug fluorescence intensity measured by FACS between vehicle and BafA1-treated conditions. Data represent n = 3 biological replicates, each with 2–3 technical replicates. Statistical significance was assessed using two-way ANOVA followed by Sidak's multiple comparison test.

6. The authors need to add a section presenting the limitations of the study. These include those indicated above and some identified in the discussion but they should be described in more detail. These would include only 3 samples, a limited spectrum of PARPi, use of slices and cell lines rather than samples from treated patients, lack of correlation between the characteristics in the models and patient outcomes which relates directly to the paragraph on clinical relevance and others.

We thank the reviewer for raising this oversight, and agree that a limitations section would be beneficial. This has now been added and covers the factors raised by Reviewer 3 as well as additional points raised by Reviewers 1 and 2. This can be found in **lines 537-554**.

Minor issues

1. The manuscript refers to Figure 6g to demonstrate reduced intracellular drug levels in the presence of bafilomycin. However, Figure 6g appears to be missing from the current version of the manuscript. Based on the content, it seems likely that the authors intended to refer to Figure 6f instead. Please review the figures carefully, and if Figure 6g is indeed missing, ensure it is included in the revised submission.

We thank the reviewer for pointing out this error and apologise for missing it. The text has been updated and corrected references to Figure 6 d, e and f applied. This can be found in lines 427-437.

Reviewer #4 (Remarks on code availability):

There were insufficient README files and instructions for running the code. Additionally, the files are restricted to users with special access. For reproducibility, the authors should make all code publicly available.

We thank the reviewer for feedback on our code accessibility. We have now added README files with much clearer instructions which are available at <https://bit.ly/3K3InPV>. The files are currently restricted so that only reviewers can access them, since this work is unpublished, however we do intend to make everything available, with README files, via Github upon publication of this manuscript.

1. Murray, J. *et al.* Tumour cell retention of rucaparib, sustained PARP inhibition and efficacy of weekly as well as daily schedules. *Brit J Cancer* **110**, 1977–1984 (2014).

2. Yang, C. & Wang, X. Lysosome biogenesis: Regulation and functions. *J. Cell Biol.* **220**, e202102001 (2021).

3. Settembre, C. & Perera, R. M. Lysosomes as coordinators of cellular catabolism, metabolic signalling and organ physiology. *Nat. Rev. Mol. Cell Biol.* **25**, 223–245 (2024).

4. Malik, N. *et al.* Induction of lysosomal and mitochondrial biogenesis by AMPK phosphorylation of FNIP1. *Science* **380**, eabj5559 (2023).

5. Beernaert, B. *et al.* Chromosomal instability shapes the tumor microenvironment of esophageal adenocarcinoma via a cGAS–chemokine–myeloid axis. *bioRxiv* 2025.05.06.652454 (2025) doi:10.1101/2025.05.06.652454.

6. McPherson, A. *et al.* Ongoing genome doubling shapes evolvability and immunity in ovarian cancer. *Nature* **644**, 1078–1087 (2025).

7. Lv, B. *et al.* A TBK1-independent primordial function of STING in lysosomal biogenesis. *Mol. Cell* (2024) doi:10.1016/j.molcel.2024.08.026.

8. Lin, W. *et al.* Autophagy confers DNA damage repair pathways to protect the hematopoietic system from nuclear radiation injury. *Sci. Rep.* **5**, 12362 (2015).

9. Hewitt, G. & Korolchuk, V. I. Repair, Reuse, Recycle: The Expanding Role of Autophagy in Genome Maintenance. *Trends Cell Biol.* **27**, 340–351 (2017).
10. Vanzo, R. *et al.* Autophagy role(s) in response to oncogenes and DNA replication stress. *Cell Death Differ.* **27**, 1134–1153 (2020).
11. Sakai, W. *et al.* Functional Restoration of BRCA2 Protein by Secondary BRCA2 Mutations in BRCA2-Mutated Ovarian Carcinoma. *Cancer Res* **69**, 6381–6386 (2009).

REVIEWERS' COMMENTS

Reviewer #1 (Remarks to the Author):

I appreciate the authors' efforts to collect in vivo data to support the conclusions of the paper. It is unfortunate that the technical limitations were unable to shed light on the in vivo distribution of these drugs. It is apparent that without tissue slicing, the millimeter-thick tumors did not have enough oxygen delivery to the center for ex vivo staining with 24 hrs in culture. It also appears, surprisingly, that the oral dosing was not able to achieve significant uptake despite the literature precedent for delivery at these doses. Potentially transporter differences (e.g. Murray et al. 2014, BJC), reduced interstitial pH in the tumor, etc. could result in differential tumor uptake in this model.

Without the in vivo data to support a lack of tissue-level gradients, I feel this limitation should be expanded in the newly added section on limitations. It is unclear if the authors were using an orthotopic model (vs. subcutaneous), but the peripheral uptake from the IP delivery seems like it could be. The tissue-level gradients seen in this IP data actually seem to confirm the effect that I was mentioning. The high cellular uptake driven by lysosomal sequestration could be limiting the drug penetration from the surface of the tumor into the tumor core (given the apparent lack of sufficient tumor perfusion for vascular delivery in the center). Presumably, a reduction in lysosomal uptake (e.g. by BafA1, although it is not practical to do this experimentally) would reduce the signal in the tumor periphery and increase it in the tumor center. These are the tissue level (distance from blood vessels or from tumor surface) that I am focused on. This is different than the intracellular gradients that are focus of this paper. Therefore, while I really like the cellular imaging experiment showing BafA1 reduces lysosomal uptake and nuclear signal (indicating trapping doesn't prevent exposure within the same cell), it doesn't address the potential for tissue-level gradients, hence the need for the statement in the limitation. The authors mention the widespread clinical benefit for all 3 PARP inhibitors, which is why they didn't consider in vivo delivery limitations initially. I agree that these are unlikely in this particular case, but the same could be said for the cellular heterogeneity, where the drugs are successful but cellular heterogeneity can lead to differential patient responses.

Now, despite these comments, I still don't think the tissue-level gradients are an issue for this particular drug class. While tumors aren't necessarily exposed to the C_{max} plasma concentration due to blood flow limitations for small molecules, the high C_{min}/trough concentrations for these agents would prevent significant washout and long-term gradients from being established. However, these gradients could exist for other drugs (or for these drugs with different delivery vehicles), and without direct evidence for this case, I think it should be expanded in the limitation section. I think it's fine to include the statement that these gradients are unlikely to exist in this case given the high C_{min} and C_{max} concentrations, but these gradients could exist for other drugs (or delivery vehicles with these drugs) where lysosomal sequestration in the tumor periphery, for surface uptake, or perivascular cells, for vascular delivery, could limit tissue penetration. Known examples include different formulations of doxorubicin (which can lead to perivascular gradients and/or poor tumor uptake from liposomal formulations) **and hypoxia-activated drugs**. This would help put the current work in the broader context of heterogeneous drug delivery.

The other changes have addressed my other concerns and improved the clarity of the manuscript.

We are glad the majority of changes have been well received. To address the outstanding point, we have added the following to the limitations paragraph of the discussion (line 522-529): “At the cellular level, we demonstrated that lysosomal content determines nuclear rucaparib availability, independently of the extracellular concentration (Supplementary Data Fig. 9I). However, further investigation would be required to determine whether high lysosomal sequestration in perivascular cells or the tumour periphery could drive extracellular tissue-level gradients by limiting drug penetration into deeper tumour regions. While such gradients are a known challenge for other agents (such as certain formulations of doxorubicin^{42,43}, the high clinical C_{min} and C_{max} concentrations of PARP inhibitors^{44,45} may help maintain a more uniform steady-state distribution over time.”

Reviewer #2 (Remarks to the Author):

I have now assessed the revised manuscript ‘Multimodal imaging reveals a lysosomal drug reservoir that drives heterogeneous distribution of PARP inhibitors’ by Moncayo et al. The authors have put significant work into this revision and the manuscript is now in much better shape. My requests have been addressed. Some have not shown the expected results, but I appreciate the methodological limitations, which they discuss. Equally I think that addition of clarification in the discussion (lines 512-523) has greatly helped with the transparency of the work. Furthermore, the points raised by my fellow reviewers have been in many cases addressed, thus making the manuscript more complete.

Overall I am now happy to recommend this work for publication. I regard it as an important addition in the field of Mass Spectrometric imaging based pharmacokinetic analysis.

Thank you – nothing further to address.

Reviewer #3 (Remarks to the Author):

We thank the authors for the extensive experimental data and experiments performed in response to concerns. It is disappointing that the in vivo studies were not successful in testing the hypotheses presented. The authors provide a rational for this concern which is reasonable. They mitigate the concerns somewhat by additional ex vivo and in vitro studies.

There are several in vitro/ex vivo additions that add to the paper. The short term treatment in the presence and absence of BAFA, the addition of PEO1 and PEO4 and additional cell lines all add to the manuscript and mitigate the majority of the concerns.

The manuscript is thus significantly strengthened.

Thank you – nothing further to address.

Reviewer #4 (Remarks to the Author):

Thank you – nothing further to address.